# EXTENDING $\mu$P: SPECTRAL CONDITIONS FOR FEATURE LEARNING ACROSS OPTIMIZERS

## ABSTRACT

Several variations of adaptive first-order and second-order optimization methods have been proposed to accelerate and scale the training of large language models. The performance of these optimization routines is highly sensitive to the choice of hyperparameters (HPs), which are computationally expensive to tune for large-scale models. Maximal update parameterization ($\mu$P) is a set of scaling rules which aims to make the optimal HPs independent of the model size, thereby allowing the HPs tuned on a smaller (computationally cheaper) model to be transferred to train a larger, target model. Despite promising results for SGD and Adam, deriving $\mu$P for other optimizers is challenging because the underlying tensor programming approach is difficult to grasp. Building on recent work that introduced spectral conditions as an alternative to tensor programs, we propose a novel framework to derive $\mu$P for a broader class of optimizers, including AdamW, ADOPT, LAMB, Sophia, Shampoo and Muon. We implement our $\mu$P derivations on multiple benchmark models and demonstrate zero-shot learning rate transfer across increasing model width for the above optimizers. Further, we provide empirical insights into depth-scaling parameterization for these optimizers.

## 1 INTRODUCTION

Large language models (LLMs) have achieved remarkable progress in generative AI, yet their performance and reproducibility depend on many interacting factors. A key aspect of training LLMs is the optimization routine, which can become unstable as models grow in size and complexity. To improve stability and efficiency, several modifications to existing optimizers have been proposed. For example, LAMB (You et al., 2019) proposes a layer-wise adaptive optimization routine to reduce the computational time required for training deep neural networks over large mini-batches, while Sophia (Liu et al., 2023) is a light-weight second-order method which achieves faster convergence than Adam and is more robust to non-convex landscapes. Muon is another recent optimizer designed explicitly for scaling with model size (Jordan et al., 2024; Liu et al., 2025; Bernstein, 2025).

Although these recent algorithms demonstrate strong performance, the computational overhead of hyperparameter (HP) tuning poses a fundamental scalability bottleneck for training LLMs. To address this challenge, practitioners have heuristically tuned HPs on smaller models to guide the search for optimal configurations in larger models. Recent works (Yang et al., 2021; Yang & Hu, 2020) have formalized this approach by proposing a zero-shot HP transfer algorithm based on maximal update parameterization ($\mu$P), which stabilizes feature learning across different model widths. $\mu$P is implemented by carefully scaling the weights and HPs proportional to the model width, with scaling factors tailored to the specific architecture and optimization algorithm. Under $\mu$P, feature learning is stable throughout the training process and HPs are stable across increasing model width.

For the above reasons, several recent works have derived and incorporated $\mu$P for different models (Zheng et al., 2025; Thérien et al.) and optimization algorithms (Blake et al., 2025b; Ishikawa & Karakida). Fig. 1 demonstrates the increased training stability and predictability after $\mu$P is incorporated in Sophia. Fig. 1 (left) shows that the relative mean of different feature vectors remains stable across increasing model width, thereby ensuring maximal (weights not decreasing to 0) and stable (weights not diverging) feature learning under $\mu$P. Fig. 1 (middle) demonstrates zero-shot learning rate transfer across model widths where the best validation loss is obtained at learning rate 0.1 for all widths. Finally, Fig. 1 (right) demonstrates the "wider is always better" property where the training loss improves consistently with increasing model width under $\mu$P.

While $\mu$P delivers strong results, it is tedious to implement in existing large codebases and difficult to understand in practice. To address this, authors in (Yang et al., 2023a) proposed simpler spectral scaling conditions on the weight matrices that lead to the same width-independent and maximal feature learning properties of $\mu$P. This work focuses on using the more tractable spectral conditions to derive $\mu$P for a wide range of optimizers. Despite being more intuitive, using spectral conditions to derive $\mu$P is not trivial and the analysis for each adaptive optimizer is different and requires a careful study of the order-of-magnitude of the coefficient terms that scale the gradients.

Our contributions are as follows: (1) we propose a general framework to derive $\mu$P using a novel spectral scaling approach; (2) we use the proposed framework to analytically derive $\mu$P for several adaptive first and second-order optimizers (AdamW, ADOPT, LAMB, Sophia, Shampoo, Muon); (3) we implement $\mu$P for the above optimizers and validate our implementation by demonstrating zero-shot HP transfer (specifically of the optimal learning rate) across model width on benchmark LLMs (NanoGPT (Karpathy, 2022); Llama2 (Touvron et al., 2023) ); and (4) we provide an empirical study of zero-shot HP transfer across model depth for these optimizers to motivate future work.

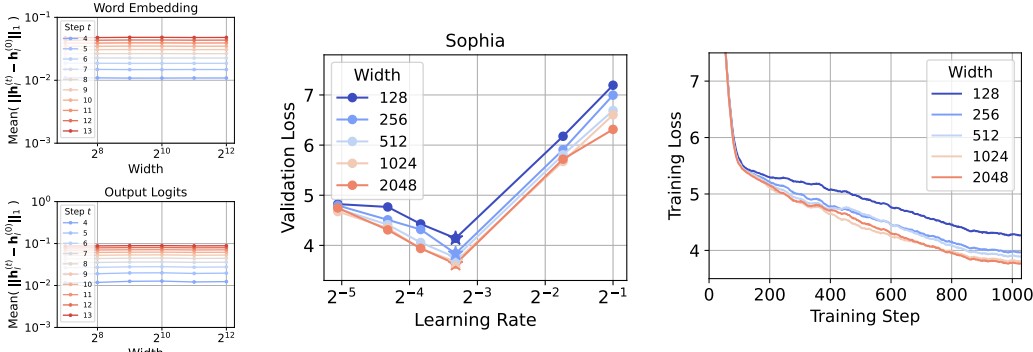

Figure 1: $\mu$P for Sophia (trained on Llama2) - Coordinate check plots for the word embedding and output logits layers (left); Zero-shot learning rate transfer across increasing model width (middle); Decreasing training loss with increasing model width (right).

## 2 PRELIMINARIES

The $l^p$−norm of a vector $\mathbf{x} \in \mathbb{R}^n$ is defined as $||\mathbf{x}||_p := \left(\sum_{i=1}^n |x_i|^p\right)^{1/p}$. For a matrix $\mathbf{A} \in \mathbb{R}^{n \times n}$, $\mathbf{A}^\alpha = \sum_i \lambda_{e_i}^\alpha \mathbf{u}_i \mathbf{u}_i^\mathrm{T}$ where $(\lambda_{e_i}, \mathbf{u}_i)$ are the $i$−th eigen pair. The spectral norm of a matrix $\mathbf{A} \in \mathbb{R}^{m \times n}$ is defined as $||\mathbf{A}||_* := \max_{\mathbf{x} \in \mathbb{R}^n \setminus \{\mathbf{0}\}} \frac{||\mathbf{A}\mathbf{x}||_2}{||\mathbf{x}||_2}$, and the Frobenius norm is defined as $||\mathbf{A}||_\mathrm{F} := \sqrt{\sum_{i=1}^m \sum_{j=1}^n |\mathbf{A}_{i,j}|^2}$ (Strang, 2012; Meyer, 2023). If $r$ denotes the rank of matrix $\mathbf{A}$, then $||\mathbf{A}||_* \leq ||\mathbf{A}||_\mathrm{F} \leq \sqrt{r}||\mathbf{A}||_*$. If a matrix $\mathbf{A} \in \mathbb{R}^{m \times n}$ can be written as an outer product of some vectors $\mathbf{u} \in \mathbb{R}^m$ and $\mathbf{v} \in \mathbb{R}^n$, that is, $\mathbf{A} = \mathbf{u}\mathbf{v}^\mathrm{T}$ then matrix $\mathbf{A}$ is a rank one matrix and

$$||\mathbf{A}||_* = ||\mathbf{A}||_\mathrm{F} = ||\mathbf{u}||_2 \cdot ||\mathbf{v}||_2. \tag{1}$$

For any symmetric matrix, the spectral norm is equal to the absolute value of the maximum eigen value. Therefore, for $p \in \mathbb{R}$, for a symmetric rank one matrix $\mathbf{A} = \mathbf{u}\mathbf{u}^\mathrm{T} \in \mathbb{R}^{n \times n}$,

$$||\mathbf{A}^p||_* = ||\mathbf{A}||_*^p. \tag{2}$$

A sequence of random vectors $\{\mathbf{x}_i \in \mathbb{R}^n\}_{i=1}^\infty$ is said to have $\Theta(n^\alpha)$-sized coordinates if there exists constants $A, B$ such that $An^\alpha \leq \sqrt{\frac{||\mathbf{x}_i||_2^2}{n}} \leq Bn^\alpha$ for all $i$, and for sufficiently large $n$.

## 3 BACKGROUND

In Sections 3, 4 and Appendix A, $\mu$P is derived for a linear MLP trained with a batch size of 1, similar to the model used in (Yang et al., 2023a). Let us consider an MLP with $L$ layers. Let $\mathbf{x} \in \mathbb{R}^{n_0}$ denote the input vector and $\mathbf{W}_l \in \mathbb{R}^{n_l \times n_{l-1}}$ denote the weight matrix for the $l$−th layer of the model. Then the feature vector $\mathbf{h}_l \in \mathbb{R}^{n_l}$ for the input $\mathbf{x}$ is given as

$$\mathbf{h}_l(\mathbf{x}) = \mathbf{W}_l \mathbf{h}_{l-1}(\mathbf{x}), \qquad \forall l = 1, 2, \ldots, L \tag{3}$$

where $\mathbf{h}_0(\mathbf{x}) = \mathbf{x}$. Let $\mathcal{L} = g(\mathbf{h}_L(\mathbf{x}), \mathbf{y})$ denote the loss, where $g : \mathbb{R}^{n_0} \times \mathbb{R}^{n_L} \to \mathbb{R}$ is a loss function, $\mathbf{y} \in \mathbb{R}^{n_L}$ is the target vector corresponding to the input $\mathbf{x}$ and $\mathbf{h}_L(\mathbf{x}) \in \mathbb{R}^{n_L}$ is the output vector returned by the MLP. After one step of training, the change in the weight matrices is typically a function, $\Psi(\cdot)$, of the history of the gradients. Then, the change in weights from time instant $t$ to $t + 1$ can be written using the following generic update rule,

$$\mathbf{W}_l^{(t+1)} = \mathbf{W}_l^{(t)} - \eta^{(t+1)} \Psi(\{\nabla_{\mathbf{W}_l^{(i)}} \mathcal{L}\}_{i=1}^t) \tag{4}$$

where $\eta^{(t+1)}$ is the learning rate at time instant $t + 1$. We specify the forms of $\Psi(\cdot)$ for different optimizers in Table 1. To reduce cumbersome notation, we omit time indices in the remaining sections unless their inclusion is necessary for clarity. This will not affect the derivation of $\mu$P as it is sufficient to analyze a single step of rule (4) to determine the correct scaling laws (Yang et al., 2021; Blake et al., 2025a). Using eqs. (3) and (4) the change in weights and feature vectors for any layer $l$, after one training step can be written as

$$\Delta \mathbf{W}_l = -\eta \Psi(\{\nabla_{\mathbf{W}_l} \mathcal{L}\}) \quad \text{and} \quad \Delta \mathbf{h}_l(\mathbf{x}) = \Delta \mathbf{W}_l \mathbf{h}_{l-1}(\mathbf{x}) + \Delta \mathbf{W}_l \Delta \mathbf{h}_{l-1}(\mathbf{x}) + \mathbf{W}_l \Delta \mathbf{h}_{l-1}(\mathbf{x}).$$

| Optimizer | $\Psi(\cdot)$ |
|---|---|
| AdamW / ADOPT | $\dfrac{\hat{\mathbf{m}}^{(t)}}{\sqrt{\hat{\mathbf{v}}^{(t)}} + \epsilon} + \lambda \mathbf{W}_l^{(t)}$ |
| Sophia | $\text{clip}\left(\dfrac{\mathbf{m}^{(t)}}{\max\{\gamma \mathbf{h}^{(t)}, \epsilon\}}, 1\right) + \lambda \mathbf{W}_l^{(t)}$ |
| LAMB | $\dfrac{\phi(\|\mathbf{W}_l^{(t)}\|_{\mathrm{F}})}{\|\mathbf{r}_l^{(t)} + \lambda \mathbf{W}_l^{(t)}\|_{\mathrm{F}}} \left(\mathbf{r}_l^{(t)} + \lambda \mathbf{W}_l^{(t)}\right)$ |
| Shampoo | $(\mathbf{L}^{(t)})^{-1/4} \nabla_{\mathbf{W}_l^{(t)}} \mathcal{L} \, (\mathbf{R}^{(t)})^{-1/4}$ |
| Muon | $\sqrt{\dfrac{n_l}{n_{l-1}}} \mathbf{O}_l^{(t)}$ |

Table 1: Values of $\Psi(\cdot)$ for different optimizers. Auxiliary variables are defined in Section 4 and Appendix A.

## 3.1 MAXIMAL UPDATE PARAMETRIZATION ( $\mu$P )

Authors in (Yang & Hu, 2020; Yang et al., 2021) proposed $\mu$P to ensure that overparameterized models do not learn trivial features, or that the feature values do not blow up with increasing model width. In practice, $\mu$P is implemented via the $abc$-parameterization (Yang & Hu, 2020) which ensures that the MLP weights, their initial variance and the learning rate are appropriately scaled with respect to the model width. In Yang & Hu (2020), the $abc$-parameterization was introduced for MLPs where the hidden layers have the same width, that is, $n_{l-1} = n_l = n$ for $l = 2, \ldots, L - 1$. For simplicity, it was assumed that the inputs and outputs are scalars. Then, for each layer, the set of parameters $\{a_l, b_l\}_{l=1}^L \cup \{c\}$ comprise the $abc$-parameterization to

1. Initialize and scale weight matrices at every layer as $\mathbf{W}_l = n^{-a_l}[\mathbf{w}_l^{(i,j)}]$, where $\mathbf{w}_l^{(i,j)} \sim \mathcal{N}(0, n^{-2b_l} \sigma^2)$
2. Scale the learning rate such that $\Delta \mathbf{W}_l = -\eta \, n^{-c} \, \Psi(\{\nabla_{\mathbf{W}_l} \mathcal{L}\})$

where the scale of initial variance, $\sigma^2$, and the learning rate, $\eta$, is assumed to be width-independent. As emphasized in Section 1, the theoretical principles behind $\mu$P can be difficult to grasp. Recognizing these challenges, (Yang et al., 2023a) provided the following equivalent conditions for $\mu$P

$$\|\mathbf{h}_l(\mathbf{x})\|_2 = \Theta(\sqrt{n_l}) \quad \text{and} \quad \|\Delta \mathbf{h}_l\|_2 = \Theta(\sqrt{n_l}), \quad \text{for} \quad l = 1, 2, \ldots, L - 1. \tag{C.1.}$$

The above conditions concisely represent the requirements of $\mu$P.

## 3.2 SPECTRAL CONDITIONS FOR FEATURE LEARNING

In (Yang et al., 2023a), the authors futher argued that conditions (C.1.) can be ensured by the following *spectral scaling conditions* on the weight matrices and their one step update,

$$\|\mathbf{W}_l\|_* = \Theta\left(\sqrt{\frac{n_l}{n_{l-1}}}\right) \quad \text{and} \quad \|\Delta \mathbf{W}_l\|_* = \Theta\left(\sqrt{\frac{n_l}{n_{l-1}}}\right), \quad \text{for} \quad l = 1, 2, \ldots, L. \tag{C.2.}$$

The above spectral scaling conditions hold for any optimizer, and in the next section we present a framework to derive $\mu$P for any arbitrary optimizer using conditions (C.2.).

### 3.3 THEORY TO PRACTICE

While the $\mu$P scalings in Table 2 are derived for the model described in the beginning of Section 3, empirical results in Fig. 2 and Fig. 3 show that the derivations also hold for more practical, complex models. This section lists the assumptions required for the derived scalings to hold in practice .

We first need to justify that deriving $\mu$P based on one time step analysis recursively yields the same scaling in the following time steps. This holds if the order of magnitude of the norms remain the same after the updates are performed, and this is formalized in Assumption 1. Note that violating Assumption 1 will require exact cancellation which is rare to observe in practice and can be easily avoided by adding small randomness to the learning rate (Yang et al., 2023a).

**Assumption 1** *The weight updates do not cancel initial quantities.*

$$||\mathbf{W}_l + \Delta\mathbf{W}_l||_* = \Theta(||\mathbf{W}_l||_* + ||\Delta\mathbf{W}_l||_*)$$
$$||\mathbf{h}_l(\mathbf{x}) + \Delta\mathbf{h}_l(\mathbf{x})||_2 = \Theta(||\mathbf{h}_l(\mathbf{x})||_2 + ||\Delta\mathbf{h}_l(\mathbf{x})||_2).$$

In practice, nonlinear activation functions, $\phi(\cdot)$, act on incoming feature vectors from the previous layer, thereby changing (3) to $\mathbf{h}_l(\mathbf{x}) = \mathbf{W}_l\phi(\mathbf{h}_{l-1}(\mathbf{x}))$. Our analysis directly translates to activation functions that preserve the order of magnitude of the inputs, as formalized in Assumption 2, and this phenomenon is observed for most commonly used activations which are designed to prevent the outputs from diverging or vanishing to 0. Additionally, Assumption 2 also holds for most transformer layers where the activation functions are preceded by layer normalization, because the normalization maps the vectors to nonnegative constants.

**Assumption 2** *If a nonlinear activation function $\phi(\cdot)$ is added to each layer of the MLP, then*

$$||\phi(\mathbf{h}_l(\mathbf{x}))||_2 = \Theta(||\mathbf{h}_l(\mathbf{x})||_2).$$

Finally, we require mild assumptions on the batch size, as stated in Assumption 3. Mathematically, Assumption 3 is required to ensure that the sub-multiplicative property of norms doesn't result in a loose bound for the derivations in Section 4 to hold in practice. Intuitively, Assumption 3 holds if the update matrix $\Delta\mathbf{W}_l$ has a low rank even for large batch sizes. We refer the reader to (Yang et al., 2023a, Figure 1) for empirical observations of low-rank behavior of update matrices.

**Assumption 3** *The batch size, $B$, is fixed and independent of the width, that is, $B = \Theta(1)$. If $i$ denotes the index of a training sample in the batch then,*

$$\|\Delta\mathbf{W}_l\mathbf{h}_l(\mathbf{x}_i)\|_2 = \Theta\left(\left\|\frac{1}{B}\Delta\mathbf{W}_l^{(i)}\mathbf{h}_l(\mathbf{x}_i)\right\|_2\right).$$

**Remark 1** *We note that Assumption 3 constitutes a limitation of $\mu$P as it implies a fixed batch size across model width. This is often suboptimal, as the critical batch size typically increases with model size (McCandlish et al., 2018; Kaplan et al., 2020). In practice, however, this can be mitigated by first tuning the smaller proxy model with a fixed batch size $B$. When transferring to larger models, one can increase the batch size to improve parallelization efficiency, provided the learning rate is adjusted accordingly. Standard heuristics for this adjustment include the linear scaling rule (Goyal et al., 2017) or square root scaling (Krizhevsky, 2014; Hoffer et al., 2017).*

## 4 DERIVING $\mu$P USING SPECTRAL SCALING CONDITIONS

As discussed in Section 3.1, deriving $\mu$P for a particular model and optimizer boils down to determining the scaling parameters in $abc$-parameterization, or an equivalent form. We propose a framework which only utilizes the spectral scaling conditions (C.2.) to derive the $abc$-parameterization. The typical approach to derive $\mu$P is to determine the proper scaling factors for a one step gradient update, and then argue recursively that for stable input vectors under $\mu$P, the output vectors are also stable, independent of the time (Assumption 1).

## 4.1 GENERIC FRAMEWORK

**Scaling of Model Weights and Initial Variance:**

The scaling factors for the model weights and their initial variance, that is, akin to parameters $\{a_l, b_l\}_{l=1}^L$ in the $abc$-parameterization, can be computed by satisfying the condition on $||\mathbf{W}_l||_*$ in (C.2.). More rigorously, let us define the model weights as $\mathbf{W}_l = \sigma_l \tilde{\mathbf{W}}_l \in \mathbb{R}^{n_l \times n_{l-1}}$ where the elements of $\tilde{\mathbf{W}}_l$ are sampled from some initial distribution with scaled variance, $n^{-2b_l}\sigma^2$. For ease of theoretical analysis, we fix $b_l = 0$ for all layers. Then, $||\mathbf{W}_l||_* = \sigma_l||\tilde{\mathbf{W}}_l||_*$. Since $||\tilde{\mathbf{W}}_l||_*$ is a random matrix with unit variance, existing results in random matrix theory can be leveraged to deduce the scaling of the spectral norm in terms of matrix dimensions (Rudelson & Vershynin, 2010) Vershynin (2018). Then, $\sigma_l$ can be computed by equating $\sigma_l||\tilde{\mathbf{W}}_l||_* = \Theta\left(\sqrt{n_l/n_{l-1}}\right)$.

**Scaling of Learning Rate:**

The scaling factor for the learning rate, akin to parameter $c$ in $abc$-parameterization, is computed by satisfying the condition on $||\Delta\mathbf{W}_l||_*$ in (C.2.). This implies that the generic update rule in eq. (4) should be equated as,

$$||\Delta\mathbf{W}_l||_* = \eta(n_l)^{-c_1}(n_{l-1})^{-c_2}||\Psi\left(\nabla_{\mathbf{W}_l}\mathcal{L}\right)||_* = \Theta\left(\sqrt{\frac{n_l}{n_{l-1}}}\right), \quad (5)$$

where the scaling constants $c_1$ and $c_2$ are determined based on the exact nature of $\Psi(\cdot)$.

|  | Input Weights | Output Weights | Hidden Weights |
|---|---|---|---|
| Init. Var. | $1\left(\frac{1}{n_{l-1}}\right)$ | $1\left(\frac{1}{n_{l-1}^2}\right)$ | $1\left(\frac{1}{n_{l-1}}\right)$ |
| Multiplier | $\frac{1}{\sqrt{n_{l-1}}}$ (1) | $\frac{1}{n_{l-1}}$ (1) | $\frac{1}{\sqrt{n_{l-1}}}$ (1) |
| AdamW / ADOPT | 1 (1) | $\frac{1}{n_{l-1}}\left(\frac{1}{n_{l-1}}\right)$ | $\frac{1}{n_{l-1}}\left(\frac{1}{n_{l-1}}\right)$ |
| Sophia LR | $1\,(-)$ | $\frac{1}{n_{l-1}}\,(-)$ | $\frac{1}{n_{l-1}}\,(-)$ |
| LAMB LR | $1\,(-)$ | $1\,(-)$ | $1\,(-)$ |
| Shampoo LR | $\sqrt{n_l}\,(-)$ | $\frac{1}{\sqrt{n_{l-1}}}\,(-)$ | $\sqrt{\frac{n_l}{n_{l-1}}}\,(-)$ |
| Muon LR (designed for hidden layers only) | NA | NA | $1\,(-)$ |

Table 2: Comparison of $\mu$P from spectral conditions (black) vs. tensor programs (Yang et al., 2021, Table 3) (red).

**Discussion:** Observe that the scaling of model weights and initial variance is only dependent on the model architecture, not the optimization routine. Therefore, in the rest of this work we use the linear MLP described in Section 3 as our fixed model architecture and assume that the weights are initialized using standard normal distribution. Since the spectral norm of a random matrix with unit variance scales $\approx (\sqrt{n_l} + \sqrt{n_{l-1}})$, the appropriate scaling factor is computed to be $\sigma_l = \Theta\left(\frac{1}{\sqrt{n_{l-1}}}\min\left\{1, \sqrt{\frac{n_l}{n_{l-1}}}\right\}\right)$ (Yang et al., 2023a). Note that the initial variance is fixed as 1 for the ease of theoretical analysis. In practice, to increase numerical stability, the variance can be set to $\sigma_l^2$ while the weight multiplier can be fixed to 1, for normal distribution.

Further, observe that eq. (5) computes separate scaling factors for the input and output dimensions of the weight matrices, that is, using spectral scaling conditions to derive $\mu$P allows us to collectively analyze the different types of layers (input, output and hidden layers). We recommend first determining the scaling factors $c_1$ and $c_2$ by removing additional HPs, such as weight-decay, epsilon for numerical stability etc., from the update rule because they typically do not have a comparable order of magnitude to other terms. In case of low-precision training (Blake et al., 2025a), these HPs can be scaled after $c_1$ and $c_2$ have been computed, as demonstrated at the end of Section 4.2.

Finally, we want to highlight that while there is no difference in the correctness and rigor of using either a tensor programming approach or the proposed spectral scaling approach, the latter is more intuitive and therefore, makes it easier to adopt and reason about $\mu$P for a wide class of optimizers.

Additionally, the rich literature on spectral norms and their properties can be leveraged to analyze different adaptive optimization routines, as will be demonstrated in the following sections.

In Section 4.2, we first demonstrate how to utilize the above framework by deriving $\mu$P for AdamW, and corroborate our results with the $\mu$P scalings reported in literature (Yang et al., 2021). We then derive $\mu$P for optimizers - ADOPT, LAMB, Sophia, Shampoo and Muon, which have shown promising results for training LLMs. Our results are summarized in Table 2 and in Result 4.1. Figs. 2 and 3 demonstrate zero-shot learning rate transfer across model widths for different optimizers, under the derived $\mu$P scalings.

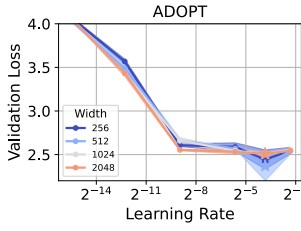 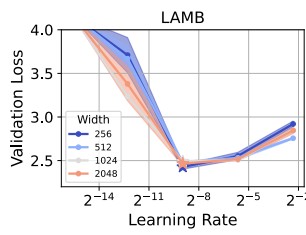

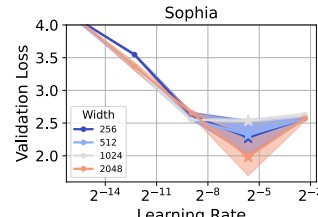 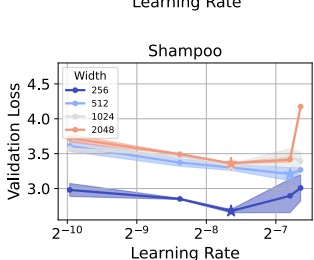

Figure 2: (NanoGPT) Mean validation loss for increasing model width and different learning rates across four optimizers: ADOPT (top left), LAMB (top right), Sophia (bottom left), and Shampoo (bottom right). The plots demonstrate zero-shot learning rate transfer under $\mu$P (Table 2).

**Result:** Under standing assumptions, for a linear MLP with $L$ layers, if the weight matrices $\mathbf{W}_l = \sigma_l \tilde{\mathbf{W}}_l$, $l = 1, 2, \ldots L$ are initialized as $\tilde{\mathbf{W}}_{i,j} \sim \mathcal{N}(0, 1)$, then the spectral conditions (C.2.) are satisfied for AdamW, ADOPT and Sophia if

$$\sigma_l = \Theta\left(\frac{1}{\sqrt{n_{l-1}}} \min\left\{1, \sqrt{\frac{n_l}{n_{l-1}}}\right\}\right); \qquad \eta = \Theta\left(\frac{1}{n_{l-1}}\right),$$

for LAMB and Muon if

$$\sigma_l = \Theta\left(\frac{1}{\sqrt{n_{l-1}}} \min\left\{1, \sqrt{\frac{n_l}{n_{l-1}}}\right\}\right); \qquad \eta = \Theta\left(1\right),$$

and for Shampoo if

$$\sigma_l = \Theta\left(\frac{1}{\sqrt{n_{l-1}}} \min\left\{1, \sqrt{\frac{n_l}{n_{l-1}}}\right\}\right); \qquad \eta = \Theta\left(\sqrt{\frac{n_l}{n_{l-1}}}\right),$$

where $n_{l-1} = 1$ for input weights and $n_l = 1$ for output weights.

**Remark 2** *For a linear MLP trained with a batch size of $1$, the gradient matrix is a rank one matrix because it can be written as an outer product of two vectors, $\nabla_{\mathbf{W}_l}\mathcal{L} = \nabla_{\mathbf{h}_l}\mathcal{L} \cdot \mathbf{h}_{l-1}^{\mathrm{T}}$. Therefore, $||\nabla_{\mathbf{W}_l}\mathcal{L}||_* = ||\nabla_{\mathbf{W}_l}\mathcal{L}||_{\mathrm{F}}$ from property (1). (See discussion in (Yang et al., 2023a, p. 9))*

**Remark 3** *For a linear MLP trained with a batch size of $1$, it can be shown using first order Taylor series expansion that $||\nabla_{\mathbf{W}_l}\mathcal{L}||_* = \Theta(\sqrt{\frac{n_{l-1}}{n_l}})$ (Yang et al., 2023a, p. 9). Further, since $\nabla_{\mathbf{W}_l}\mathcal{L}$ is a rank one matrix, $||\nabla_{\mathbf{W}_l}\mathcal{L}||_* = ||\nabla_{\mathbf{h}_l}\mathcal{L}||_2||\mathbf{h}_{l-1}||_2 = ||\nabla_{\mathbf{h}_l}\mathcal{L}||_2\Theta(\sqrt{n_{l-1}})$, using property (1) and condition (C.1.). Then, $||\nabla_{\mathbf{h}_l}\mathcal{L}||_2 = \Theta(1/\sqrt{n_l})$.*

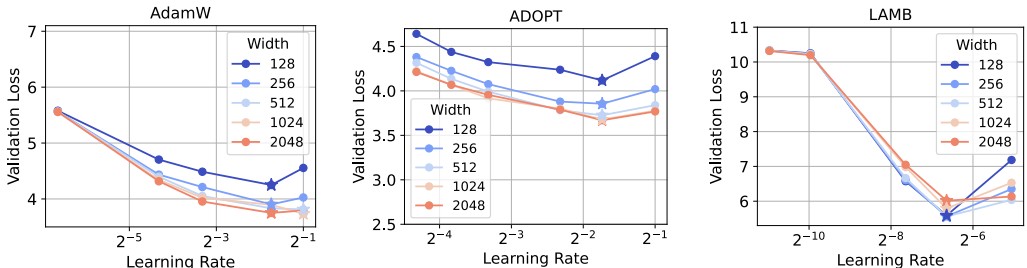

Figure 3: (Llama2) Validation loss for increasing model width and different learning rates across three optimizers: AdamW (left), ADOPT (middle), and LAMB (right). The plots demonstrate zero-shot learning rate transfer under $\mu$P (Table 2).

### 4.2 $\mu$P FOR ADAMW

Recall the update rule for AdamW (Loshchilov & Hutter, 2017),

$$\mathbf{W}_l^{(t+1)} = \mathbf{W}_l^{(t)} - \eta^{(t+1)} \left( \frac{\hat{\mathbf{m}}^{(t)}}{\sqrt{\hat{\mathbf{v}}^{(t)}} + \epsilon} + \lambda \mathbf{W}_l^{(t)} \right) \quad \text{(AdamW)}$$

where $\quad \hat{\mathbf{m}}^{(t)} = \dfrac{\mathbf{m}^{(t)}}{(1 - \beta_1^t)} = \dfrac{1}{(1 - \beta_1^t)} \left[ \beta_1 \mathbf{m}^{(t-1)} + (1 - \beta_1) \nabla_{\mathbf{W}_l^{(t)}} \mathcal{L} \right] \quad ; \quad \mathbf{m}^{(0)} = 0$

$$\hat{\mathbf{v}}^{(t)} = \frac{\mathbf{v}^{(t)}}{(1 - \beta_2^t)} = \frac{1}{(1 - \beta_2^t)} \left[ \beta_2 \mathbf{v}^{(t-1)} + (1 - \beta_2)(\nabla_{\mathbf{W}_l^{(t)}} \mathcal{L})^2 \right] \quad ; \quad \mathbf{v}^{(0)} = 0$$

From the spectral scaling condition in eq. (5), we need to find $c_1, c_2 \in \mathbb{R}$ such that

$$||\Delta \mathbf{W}_l||_* = \eta(n_l)^{-c_1}(n_{l-1})^{-c_2} \left\| \frac{\hat{\mathbf{m}}}{\sqrt{\hat{\mathbf{v}}} + \epsilon} + \lambda \mathbf{W}_l \right\|_* = \Theta \left( \sqrt{\frac{n_l}{n_{l-1}}} \right). \quad (6)$$

Similar to previous works, we first analyze AdamW for $\beta_1 = \beta_2 = \epsilon = 0$. Then, the above update rule reduces to signSGD (Bernstein et al., 2018). Additionally, since the gradient term dominates the weight decay term, we ignore the latter because we are only concerned with an order-of-magnitude calculation. Therefore, (6) reduces to

$$||\Delta \mathbf{W}_l||_* = \eta(n_l)^{-c_1}(n_{l-1})^{-c_2}||\text{sign}(\nabla_{\mathbf{W}_l} \mathcal{L})||_* \approx \eta(n_l)^{-c_1}(n_{l-1})^{-c_2}||\text{sign}(\nabla_{\mathbf{W}_l} \mathcal{L})||_F$$

where the last equation follows from Remark 2. From the definition of the Frobenius norm, we have $||\mathbf{1}_{n_l \times n_{l-1}}||_F^2 = \sum_{i=1}^{n_l} \sum_{j=i}^{n_{l-1}} 1 = n_l n_{l-1}$. This gives

$$||\Delta \mathbf{W}_l||_* = \eta(n_l)^{-c_1}(n_{l-1})^{-c_2} \Theta \left( \sqrt{n_l n_{l-1}} \right) = \Theta \left( n_l^{1/2 - c_1} n_{l-1}^{1/2 - c_2} \right). \quad (7)$$

By fixing $c_1 = 0$ and $c_2 = 1$, the spectral scaling condition in eq.(5) is satisfied. Therefore, the learning rate for AdamW should be scaled by a factor of $1/n_{l-1}$. Observe that this scaling is consistent with the $\mu$P derived using the tensor programming approach (Yang et al., 2021, Table 3), and this equivalence is highlighted in Table 2. Fig. 4 further validates our derivation via the coordinate check plots and the "wider is better" phenomenon observed in the plot on the right. Since the update rule of ADOPT is similar to AdamW, we discuss $\mu$P for ADOPT in Appendix A.

**Scaling of Momentum, Adaptive Noise, and Weight Decay terms:**

Typically, HPs like $\beta_1$ and $\beta_2$ are width-independent and have $\Theta(1)$ order of magnitude. Thus, these parameters are not dominant when analyzing the momentum terms and do not require separate scaling rules. Similarly, the adaptive noise term $\epsilon$ requires no scaling if it is fixed at a very small value. However, empirical studies show that $\epsilon$ may affect the performance of $\mu$P under certain training regimes (Blake et al., 2025a; Dey et al., 2025). In such cases the scaling law for $\epsilon$ can be derived as follows. From (AdamW), we observe that for the above scaling law to hold, the spectral norm of $\epsilon$ should have the same order of magnitude as the spectral norm of $\sqrt{\hat{v}}$. Now,

$||\sqrt{\hat{v}}||_* = ||\nabla_{\mathbf{W}_l}\mathcal{L}||_* = \Theta(\sqrt{n_{l-1}/n_l})$ and $||\epsilon\mathbf{1}_{n_l \times n_{l-1}}||_* \approx \epsilon||\mathbf{1}_{n_l \times n_{l-1}}||_{\mathrm{F}} = \epsilon\Theta(\sqrt{n_l n_{l-1}})$. Therefore, a factor of $\frac{1}{n_l}$ scales $\epsilon$ to the appropriate order of magnitude.

On the other hand, for the derived $\mu$P scaling to hold for (AdamW), the spectral norm of the weight decay term, $||\lambda\mathbf{W}_l||_*$, must have the same order of magnitude as the spectral norm of the gradient term, which is $\Theta(\sqrt{n_l n_{l-1}})$. Since, $||\lambda\mathbf{W}_l||_* = \lambda||\mathbf{W}_l||_* = \lambda\Theta(\sqrt{n_l/n_{l-1}})$, where the last equality follows from condition (C.2.), then $\lambda$ should be scaled by a factor of $n_{l-1}$. The above results are consistent with Table 1 in (Dey et al., 2025).

### 4.3 $\mu$P FOR LAMB

Recall the update rule for LAMB (You et al., 2019),

$$\mathbf{W}_l^{(t+1)} = \mathbf{W}_l^{(t)} - \eta^{(t+1)} \frac{\phi(||\mathbf{W}_l^{(t)}||_{\mathrm{F}})}{||\mathbf{r}_l^{(t)} + \lambda\mathbf{W}_l^{(t)}||_{\mathrm{F}}} \left(\mathbf{r}_l^{(t)} + \lambda\mathbf{W}_l^{(t)}\right) \tag{LAMB}$$

where $\mathbf{r}_l^{(t)} = \frac{\hat{\mathbf{m}}^{(t)}}{\sqrt{\hat{\mathbf{v}}^{(t)}}+\epsilon}$. In (LAMB), the gradient in each layer of the model is scaled by terms of orders $\frac{||\mathbf{W}_l||_{\mathrm{F}}}{||\mathbf{r}_l + \lambda\mathbf{W}_l||_{\mathrm{F}}}$. From condition (C.2.), we know $||\mathbf{W}_l||_{\mathrm{F}} \approx ||\mathbf{W}_l||_* = \Theta\left(\sqrt{\frac{n_l}{n_{l-1}}}\right)$. Observe that the term in the denominator is the update rule for (AdamW) and we can use the result in (7) to determine its order of magnitude. Therefore,

$$||\mathbf{r}_l + \lambda\mathbf{W}_l||_{\mathrm{F}} = \Theta\left(\sqrt{n_l n_{l-1}}\right) \qquad \text{and} \qquad \frac{||\mathbf{W}_l||_{\mathrm{F}}}{||\mathbf{r}_l + \lambda\mathbf{W}_l||_{\mathrm{F}}} = \Theta\left(\frac{1}{n_{l-1}}\right). \tag{8}$$

Then, from the spectral scaling condition in eq. (5), we need to find $c_1, c_2 \in \mathbb{R}$ such that

$$||\Delta\mathbf{W}||_* \approx \eta(n_l)^{-c_1}(n_{l-1})^{-c_2}\Theta\left(\frac{1}{n_{l-1}}\right)||\mathbf{r}_l + \lambda\mathbf{W}_l||_{\mathrm{F}}$$

$$= \eta(n_l)^{-c_1}(n_{l-1})^{-c_2}\Theta\left(\frac{1}{n_{l-1}}\right)\Theta\left(\sqrt{n_l n_{l-1}}\right)$$

$$= \eta(n_l)^{-c_1}(n_{l-1})^{-c_2}\Theta\left(\sqrt{\frac{n_l}{n_{l-1}}}\right)$$

where the second equality follows using the same reasoning as for AdamW. Then condition (5) holds if $c_1 = c_2 = 0$. Note that by invoking result (7) from AdamW's analysis to determine the order of magnitude of $||\mathbf{r}_l + \lambda\mathbf{W}_l||_{\mathrm{F}}$ in (8), we implicitly assume that the HPs $\lambda$ and $\epsilon$ have been appropriately scaled following the analysis in Section 4.2. Therefore, the HPs in (LAMB) follow the same scaling rule as (AdamW).

**Insight 1** *The above derivation suggests that the update rule for LAMB is implicitly independent of width scaling. Intuitively, this result holds because the layerwise gradient scaling in (LAMB) causes the effective learning rate to be different for each layer.*

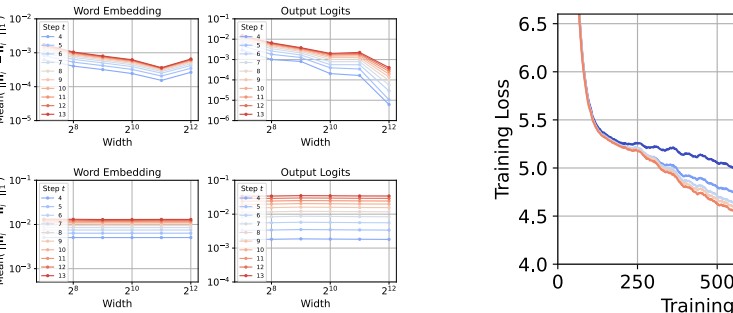

Figure 4: (Llama2 model) AdamW optimizer - Coordinate check plots under standard parameterization (top left) and under $\mu$P (bottom left) for the word embedding and output logits layers; Decreasing training loss with increasing model width under $\mu$P (right).

### 4.4 $\mu$P FOR SOPHIA

Recall the update rule for Sophia (Liu et al., 2023),

$$\mathbf{W}_l^{(t+1)} = \mathbf{W}_l^{(t)} - \eta^{(t+1)} \operatorname{clip}\left(\frac{\mathbf{m}^{(t)}}{\max\{\gamma \mathbf{h}^{(t)}, \epsilon\}}, 1\right) - \eta^{(t)} \lambda \mathbf{W}_l^{(t)} \qquad \text{(Sophia)}$$

where $\mathbf{h}^{(t)}$ is a momentum-based estimate of the diagonal vector of the Hessian at time $t$. From the spectral scaling condition in (5), we need to find $c_1, c_2 \in \mathbb{R}$ such that

$$||\Delta \mathbf{W}_l||_* = \eta(n_l)^{-c_1}(n_{l-1})^{-c_2} \left\| \operatorname{clip}\left(\frac{\mathbf{m}^{(t)}}{\max\{\gamma \mathbf{h}^{(t)}, \epsilon\}}, 1\right) - \lambda \mathbf{W}_l^{(t)} \right\|_* = \Theta\left(\sqrt{\frac{n_l}{n_{l-1}}}\right).$$

For analysis, we consider $\beta_1 = \beta_2 = \epsilon = 0$, and since the weight decay term is usually very small, the above weight update simplifies to

$$||\Delta \mathbf{W}_l||_* = \eta(n_l)^{-c_1}(n_{l-1})^{-c_2} \left\| \operatorname{clip}\left(\frac{\nabla_{\mathbf{w}_l}\mathcal{L}}{\gamma \nabla^2_{\mathbf{W}_l}\mathcal{L}}, 1\right) \right\|_*$$

$$\approx \eta(n_l)^{-c_1}(n_{l-1})^{-c_2} \left\| \operatorname{clip}\left(\frac{\nabla_{\mathbf{w}_l}\mathcal{L}}{\gamma |\nabla^2_{\mathbf{W}_l}\mathcal{L}|}, 1\right) \right\|_F$$

where we take the modulus in the denominator because Sophia avoids negative diagonal terms in the Hessian (thereby avoiding convergence to a saddle point; see discussion in (Liu et al., 2023, pg. 6)). Observe that the clip$(\cdot, 1)$ bounds the coordinate-wise weight updates as, $|[\Delta \mathbf{W}_l]_{i,j}| \leq 1$. Therefore, we can compute an upper bound for the Frobenius norm and get

$$||\Delta \mathbf{W}_l||_* \leq \eta(n_l)^{-c_1}(n_{l-1})^{-c_2}\frac{1}{\gamma}\Theta(\sqrt{n_l n_{l-1}}).$$

Then, eq. (5) is satisfied by fixing $c_1 = 0$ and $c_2 = 1$, resulting in the same $\mu$P scaling as AdamW. Note that the momentum terms $\beta_1$ and $\beta_2$ do not require any additional scaling because they have $\Theta(1)$, width-indepedent order of magnitude, where as the HPs $\lambda$ and $\epsilon$ follow the same scaling as the HPs of AdamW because Sophia and AdamW have the same $\mu$P scaling.

**Insight 2** *We provide an intuitive explanation for this result. Sophia uses signSGD as the default method to handle negative Hessian terms (to avoid convergence to a saddle point), thereby mirroring the analysis for AdamW for such cases. Additionally, when $\gamma = 1$, all the elements in the weight update are clipped to 1, and the upper bound holds exactly. Thus, we get the same scaling as AdamW.*

*In practice, the authors suggest to choose $\gamma$ such that $10\% - 50\%$ of the parameters are not clipped. Therefore, for each term which is not clipped, the above bound incurs an error of less than 1. However, as demonstrated in our simulations (Fig. 2), for the typical values of $\gamma$ used in practice, the $\mu$P scaling derived based on the above calculation works well.*

Fig. 1 further validates the $\mu$P derivation for Sophia via stable coordinate check plots (Fig. 1 (left)) and a consistently improving training loss across model widths (Fig. 1 (right)).

### 4.5 $\mu$P FOR SHAMPOO

Recall the update rule for Shampoo (Gupta et al., 2018),

$$\mathbf{W}_l^{(t+1)} = \mathbf{W}_l^{(t)} - \eta^{(t+1)} \left(\mathbf{L}_l^{(t)}\right)^{-1/4} \nabla_{\mathbf{W}_l}\mathcal{L} \left(\mathbf{R}_l^{(t)}\right)^{-1/4} \qquad \text{(Shampoo)}$$

where for some $\delta > 0$, $\quad \mathbf{L}_l^{(t)} = \mathbf{L}_l^{(t-1)} + \nabla_{\mathbf{W}_l}\mathcal{L} \cdot \nabla_{\mathbf{W}_l}\mathcal{L}^{\mathrm{T}} \quad ; \quad \mathbf{L}_l^{(0)} = \delta \mathbf{I} \in \mathbb{R}^{n_l \times n_l}$

$$\mathbf{R}_l^{(t)} = \mathbf{R}_l^{(t-1)} + \nabla_{\mathbf{W}_l}\mathcal{L}^{\mathrm{T}} \cdot \nabla_{\mathbf{W}_l}\mathcal{L} \quad ; \quad \mathbf{R}_l^{(0)} = \delta \mathbf{I} \in \mathbb{R}^{n_{l-1} \times n_{l-1}}$$

From the spectral scaling condition in (5), we need to find $c_1, c_2 \in \mathbb{R}$ such that

$$||\Delta \mathbf{W}_l||_* = \eta(n_l)^{-c_1}(n_{l-1})^{-c_2} \left\| \left(\mathbf{L}_l^{(t)}\right)^{-1/4} \nabla_{\mathbf{W}_l}\mathcal{L} \left(\mathbf{R}_l^{(t)}\right)^{-1/4} \right\|_* = \Theta\left(\sqrt{\frac{n_l}{n_{l-1}}}\right).$$

For one-step analysis, let $\delta = 0$. Then the above condition reduces to

$$\|\Delta\mathbf{W}_l\|_* = \eta(n_l)^{-c_1}(n_{l-1})^{-c_2}\left\|\left(\nabla_{\mathbf{W}_l}\mathcal{L}\cdot\nabla_{\mathbf{W}_l}\mathcal{L}^{\mathrm{T}}\right)^{-1/4}\nabla_{\mathbf{W}_l}\mathcal{L}\left(\nabla_{\mathbf{W}_l}\mathcal{L}^{\mathrm{T}}\cdot\nabla_{\mathbf{W}_l}\mathcal{L}\right)^{-1/4}\right\|_*$$

$$\overset{(1)}{\leq}\eta(n_l)^{-c_1}(n_{l-1})^{-c_2}\left\|\left(\nabla_{\mathbf{W}_l}\mathcal{L}\cdot\nabla_{\mathbf{W}_l}\mathcal{L}^{\mathrm{T}}\right)^{-1/4}\right\|_*\|\nabla_{\mathbf{W}_l}\mathcal{L}\|_*\left\|\left(\nabla_{\mathbf{W}_l}\mathcal{L}^{\mathrm{T}}\cdot\nabla_{\mathbf{W}_l}\mathcal{L}\right)^{-1/4}\right\|_*$$

$$\overset{(2)}{=}\eta\Theta\left((n_l)^{-c_1-\frac{1}{2}}(n_{l-1})^{-c_2+\frac{1}{2}}\right)$$
$$\left\|\left(\nabla_{\mathbf{h}_l}\mathcal{L}\cdot\mathbf{h}_{l-1}^{\mathrm{T}}\mathbf{h}_{l-1}\cdot\nabla_{\mathbf{h}_l}\mathcal{L}^{\mathrm{T}}\right)^{-1/4}\right\|_*\left\|\left(\mathbf{h}_{l-1}\cdot\nabla_{\mathbf{h}_l}\mathcal{L}^{\mathrm{T}}\nabla_{\mathbf{h}_l}\mathcal{L}\cdot\mathbf{h}_{l-1}^{\mathrm{T}}\right)^{-1/4}\right\|_*$$

$$\overset{(3)}{=}\eta\Theta\left((n_l)^{-c_1-\frac{1}{2}}(n_{l-1})^{-c_2+\frac{1}{2}}\right)$$
$$\Theta(n_{l-1}^{-1/4})\left\|\left(\nabla_{\mathbf{h}_l}\mathcal{L}\cdot\nabla_{\mathbf{h}_l}\mathcal{L}^{\mathrm{T}}\right)^{-1/4}\right\|_*\Theta(n_l^{1/4})\left\|\left(\mathbf{h}_{l-1}\cdot\mathbf{h}_{l-1}^{\mathrm{T}}\right)^{-1/4}\right\|_*$$

$$\overset{(4)}{=}\eta\Theta\left((n_l)^{-c_1-\frac{1}{4}}(n_{l-1})^{-c_2+\frac{1}{4}}\right)\|\nabla_{\mathbf{h}_l}\mathcal{L}\|_2^{-1/2}\|\mathbf{h}_{l-1}\|_2^{-1/2}$$

$$\overset{(5)}{=}\eta\Theta\left((n_l)^{-c_1-\frac{1}{4}}(n_{l-1})^{-c_2+\frac{1}{4}}\right)\Theta(n_l^{1/4})\Theta(n_{l-1}^{-1/4})\ =\ \eta\Theta\left((n_l)^{-c_1}(n_{l-1})^{-c_2}\right)$$

where (1) follows from sub-multiplicative property of matrix norms, (2) follows from Remark 3, (3) and (5) follow from condition (C.1.) and Remark 3, (4) follows from property (1) and property (2). Therefore, condition (5) is satisfied by fixing $c_1 = -1/2$ and $c_2 = 1/2$. Note that the $\delta$ HP in (Shampoo) is akin to the momentum HPs in (AdamW) and have a $\Theta(1)$ order of magnitude. Therefore, $\delta$ doesn't contribute to the calculations of $\mathbf{L}_l$ and $\mathbf{R}_l$, and it doesn't require any further scaling.

**Muon:** Muon was first introduced in (Jordan et al., 2024) and empirical results have demonstrated its scalability for LLMs (Liu et al., 2025). (Jordan et al., 2024) also showed the equivalence between Muon and Shampoo if the preconditioner accumulation is removed from (Shampoo). Therefore, the original version of Muon (Jordan et al., 2024) follows the same $\mu$P scaling as Shampoo. However, a more recent version of Muon (Bernstein, 2025) incorporates width-independent scaling of the learning rate explicitly in the update rule itself (Table 1). We analyze this version of Muon in Appendix A and show that no further scaling is required for stable feature learning. This conclusion is added to Result 4.1.

## 5 NUMERICAL RESULTS

We test and validate our derivations on the NanoGPT model (Karpathy (2022)) and the Llama2 model (Touvron et al. (2023)). As demonstrated in Figs. 2 and 3, our simulation results validate the $\mu$P derivations in Table 2 across the different optimizers. Extensive numerical results, including training settings, HP values, depth scaling studies, and validation loss values for the different optimizers and model sizes can be found in Appendix B. The simulations on NanoGPT were performed using four $A100$ GPUs of the Argonne Leadership Computing Facility's Polaris supercomputer (Leadership Computing Facility (b)), while the simulations on Llama2 were performed using 12 Intel Data Center GPU Max Series on the Aurora supercomputer (Leadership Computing Facility (a)).

## 6 CONCLUSION

We have proposed a novel framework to derive $\mu$P using spectral scaling conditions, which are more intuitive and easier to work with than the prevalent tensor programs. Using the proposed framework, we have derived $\mu$P for a wide range of adaptive, first and second-order optimizers including, AdamW, ADOPT, LAMB, Sophia, Shampoo and Muon. We have implemented $\mu$P for the above optimizers on two benchmark LLMs, and validated our implementation by demonstrating zero-shot learning rate transfer. Motivated by our depth-scaling simulations (Appendix B), we aim to develop a sound theoretical framework for depth-scaling parameterization in the future.

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

## A  DERIVING $\mu$P

### A.1  $\mu$P FOR ADOPT

Recall that the update rule for ADOPT is the same as AdamW. The key difference lies in the sequence in which the terms $\hat{\mathbf{m}}^{(t)}$ and $\hat{\mathbf{v}}^{(t)}$ are updated (Taniguchi et al. (2024)). From a theoretical perspective, this does not change the order of magnitude of the gradient function $\Psi(\{\nabla_{\mathbf{W}_l}\mathcal{L}\})$ from that of AdamW, and hence, the parameterization derived for AdamW also holds for ADOPT.

### A.2  $\mu$P FOR SHAMPOO (DETAILED)

We present a more detailed derivation for Shampoo in this section.

Recall the update rule for Shampoo (Gupta et al., 2018),

$$\mathbf{W}_l^{(t+1)} = \mathbf{W}_l^{(t)} - \eta^{(t+1)}\left(\mathbf{L}_l^{(t)}\right)^{-1/4}\nabla_{\mathbf{W}_l}\mathcal{L}\left(\mathbf{R}_l^{(t)}\right)^{-1/4} \tag{Shampoo}$$

where for some $\delta > 0$, $\quad \mathbf{L}_l^{(t)} = \mathbf{L}_l^{(t-1)} + \nabla_{\mathbf{W}_l}\mathcal{L}\cdot\nabla_{\mathbf{W}_l}\mathcal{L}^{\mathrm{T}} \quad ; \quad \mathbf{L}_l^{(0)} = \delta\mathbf{I}\in\mathbb{R}^{n_l\times n_l}$

$$\mathbf{R}_l^{(t)} = \mathbf{R}_l^{(t-1)} + \nabla_{\mathbf{W}_l}\mathcal{L}^{\mathrm{T}}\cdot\nabla_{\mathbf{W}_l}\mathcal{L} \quad ; \quad \mathbf{R}_l^{(0)} = \delta\mathbf{I}\in\mathbb{R}^{n_{l-1}\times n_{l-1}}$$

From the spectral scaling condition in (5), we need to find $c_1, c_2 \in \mathbb{R}$ such that

$$||\Delta\mathbf{W}_l||_* = \eta(n_l)^{-c_1}(n_{l-1})^{-c_2}\left\|\left(\mathbf{L}_l^{(t)}\right)^{-1/4}\nabla_{\mathbf{W}_l}\mathcal{L}\left(\mathbf{R}_l^{(t)}\right)^{-1/4}\right\|_* = \Theta\left(\sqrt{\frac{n_l}{n_{l-1}}}\right).$$

For one-step analysis, let $\delta = 0$. Then the above condition reduces to

$$||\Delta\mathbf{W}_l||_* = \eta(n_l)^{-c_1}(n_{l-1})^{-c_2}\left\|\left(\nabla_{\mathbf{W}_l}\mathcal{L}\cdot\nabla_{\mathbf{W}_l}\mathcal{L}^{\mathrm{T}}\right)^{-1/4}\nabla_{\mathbf{W}_l}\mathcal{L}\left(\nabla_{\mathbf{W}_l}\mathcal{L}^{\mathrm{T}}\cdot\nabla_{\mathbf{W}_l}\mathcal{L}\right)^{-1/4}\right\|_*$$

$$\overset{(1)}{\leq} \eta(n_l)^{-c_1}(n_{l-1})^{-c_2}\left\|\left(\nabla_{\mathbf{W}_l}\mathcal{L}\cdot\nabla_{\mathbf{W}_l}\mathcal{L}^{\mathrm{T}}\right)^{-1/4}\right\|_*\|\nabla_{\mathbf{W}_l}\mathcal{L}\|_*\left\|\left(\nabla_{\mathbf{W}_l}\mathcal{L}^{\mathrm{T}}\cdot\nabla_{\mathbf{W}_l}\mathcal{L}\right)^{-1/4}\right\|_*$$

$$\overset{(2)}{=} \eta(n_l)^{-c_1}(n_{l-1})^{-c_2}\Theta\left(\sqrt{\frac{n_{l-1}}{n_l}}\right)\left\|\left(\nabla_{\mathbf{W}_l}\mathcal{L}\cdot\nabla_{\mathbf{W}_l}\mathcal{L}^{\mathrm{T}}\right)^{-1/4}\right\|_*\left\|\left(\nabla_{\mathbf{W}_l}\mathcal{L}^{\mathrm{T}}\cdot\nabla_{\mathbf{W}_l}\mathcal{L}\right)^{-1/4}\right\|_*$$

$$= \eta\Theta\left((n_l)^{-c_1-\frac{1}{2}}(n_{l-1})^{-c_2+\frac{1}{2}}\right)$$

$$\left\|\left(\nabla_{\mathbf{h}_l}\mathcal{L}\cdot\mathbf{h}_{l-1}^{\mathrm{T}}\mathbf{h}_{l-1}\cdot\nabla_{\mathbf{h}_l}\mathcal{L}^{\mathrm{T}}\right)^{-1/4}\right\|_*\left\|\left(\mathbf{h}_{l-1}\cdot\nabla_{\mathbf{h}_l}\mathcal{L}^{\mathrm{T}}\nabla_{\mathbf{h}_l}\mathcal{L}\cdot\mathbf{h}_{l-1}^{\mathrm{T}}\right)^{-1/4}\right\|_*$$

$$= \eta\Theta\left((n_l)^{-c_1-\frac{1}{2}}(n_{l-1})^{-c_2+\frac{1}{2}}\right)$$

$$\left\|\left(\|\mathbf{h}_{l-1}\|_2^2\,\nabla_{\mathbf{h}_l}\mathcal{L}\cdot\nabla_{\mathbf{h}_l}\mathcal{L}^{\mathrm{T}}\right)^{-1/4}\right\|_*\left\|\left(\|\nabla_{\mathbf{h}_l}\mathcal{L}\|_2^2\,\mathbf{h}_{l-1}\cdot\mathbf{h}_{l-1}^{\mathrm{T}}\right)^{-1/4}\right\|_*$$

$$= \eta\Theta\left((n_l)^{-c_1-\frac{1}{2}}(n_{l-1})^{-c_2+\frac{1}{2}}\right)\|\mathbf{h}_{l-1}\|_2^{-1/2}$$

$$\left\|\left(\nabla_{\mathbf{h}_l}\mathcal{L}\cdot\nabla_{\mathbf{h}_l}\mathcal{L}^{\mathrm{T}}\right)^{-1/4}\right\|_*\|\nabla_{\mathbf{h}_l}\mathcal{L}\|_2^{-1/2}\left\|\left(\mathbf{h}_{l-1}\cdot\mathbf{h}_{l-1}^{\mathrm{T}}\right)^{-1/4}\right\|_*$$

$$\overset{(3)}{=} \eta\Theta\left((n_l)^{-c_1-\frac{1}{2}}(n_{l-1})^{-c_2+\frac{1}{2}}\right)\Theta(n_{l-1}^{-1/4})\,\|$$

$$\left(\nabla_{\mathbf{h}_l}\mathcal{L}\cdot\nabla_{\mathbf{h}_l}\mathcal{L}^{\mathrm{T}}\right)^{-1/4}\Big\|_*\Theta(n_l^{1/4})\left\|\left(\mathbf{h}_{l-1}\cdot\mathbf{h}_{l-1}^{\mathrm{T}}\right)^{-1/4}\right\|_*$$

$$= \eta\Theta\left((n_l)^{-c_1-\frac{1}{4}}(n_{l-1})^{-c_2+\frac{1}{4}}\right)\left\|\left(\nabla_{\mathbf{h}_l}\mathcal{L}\cdot\nabla_{\mathbf{h}_l}\mathcal{L}^{\mathrm{T}}\right)^{-1/4}\right\|_*\left\|\left(\mathbf{h}_{l-1}\cdot\mathbf{h}_{l-1}^{\mathrm{T}}\right)^{-1/4}\right\|_*$$

$$\overset{(4)}{=} \eta\Theta\left((n_l)^{-c_1-\frac{1}{4}}(n_{l-1})^{-c_2+\frac{1}{4}}\right)\|\nabla_{\mathbf{h}_l}\mathcal{L}\|_2^{-1/2}\|\mathbf{h}_{l-1}\|_2^{-1/2}$$

$$\overset{(5)}{=} \eta\Theta\left((n_l)^{-c_1-\frac{1}{4}}(n_{l-1})^{-c_2+\frac{1}{4}}\right)\Theta(n_l^{1/4})\Theta(n_{l-1}^{-1/4})$$

$$= \eta\Theta\left((n_l)^{-c_1}(n_{l-1})^{-c_2}\right)$$

where (1) follows from sub-multiplicative property of matrix norms, (2) follows from Remark 3, (3) and (5) follow from condition (C.1.) and Remark 3, (4) follows from property (1) and property (2). Therefore, condition (5) is satisfied by fixing $c_1 = -1/2$ and $c_2 = 1/2$.

### A.3 $\mu$P FOR MUON

Muon is one of the first optimizers to implicitly adopt a width-independent update rule by scaling the learning rate with a factor of $\left( \sqrt{\frac{n_l}{n_{l-1}}} \right)$. Therefore, intuitively, we do not expect any further scaling of the learning rate under $\mu$P. This conjecture is validated through the following analysis on the most recent version of Muon.

Recall the update rule for Muon (Bernstein, 2025; Jordan et al., 2024),

$$\mathbf{W}_l^{(t+1)} = \mathbf{W}_l^{(t)} - \eta^{(t+1)} \sqrt{\frac{n_l}{n_{l-1}}} \mathbf{O}_l^{(t)} \tag{Muon}$$

$$\text{where} \quad \mathbf{O}_l^{(t)} = \text{NewtonSchulz}(\mathbf{B}_l^{(t)})$$
$$\mathbf{B}_l^{(t)} = \mu \mathbf{B}_l^{(t-1)} + \nabla_{\mathbf{W}_l^{(t)}} \mathcal{L} \quad ; \quad \mathbf{B}_l^{(0)} = \mathbf{0}$$

From the spectral scaling condition in eq. (5), we need to find $c_1, c_2 \in \mathbb{R}$ such that

$$||\Delta \mathbf{W}_l||_* = \eta (n_l)^{-c_1} (n_{l-1})^{-c_2} \left\| \sqrt{\frac{n_l}{n_{l-1}}} \mathbf{O}_l \right\|_* = \Theta \left( \sqrt{\frac{n_l}{n_{l-1}}} \right) \tag{9}$$

In this analysis we are working directly with an orthogonal matrix $\mathbf{O}_l^{(t)} \in \mathbb{R}^{n_l \times n_{l-1}}$ and the spectral norm of an orthogonal matrix is 1 because the modulus of all its eigen values is 1 Horn & Johnson (2012).

$$||\Delta \mathbf{W}_l||_* = \eta (n_l)^{-c_1} (n_{l-1})^{-c_2} \sqrt{\frac{n_l}{n_{l-1}}} \left\| \mathbf{O}_l^{(t)} \right\|_*$$
$$= \eta (n_l)^{-c_1} (n_{l-1})^{-c_2} \sqrt{\frac{n_l}{n_{l-1}}}.$$

Then condition (5) holds if $c_1 = c_2 = 0$. Fig. 5 demonstrates the zero-shot learning rate transfer as well as the "wider is better" phenomenon for Muon.

Note that the initial implementation of Muon did not incorporate the scaling factor $\left( \sqrt{\frac{n_l}{n_{l-1}}} \right)$ in the update rule, but the proven equivalence between Muon and Shampoo leads to Muon having the same $\mu$P scaling as Shampoo (Jordan et al., 2024).

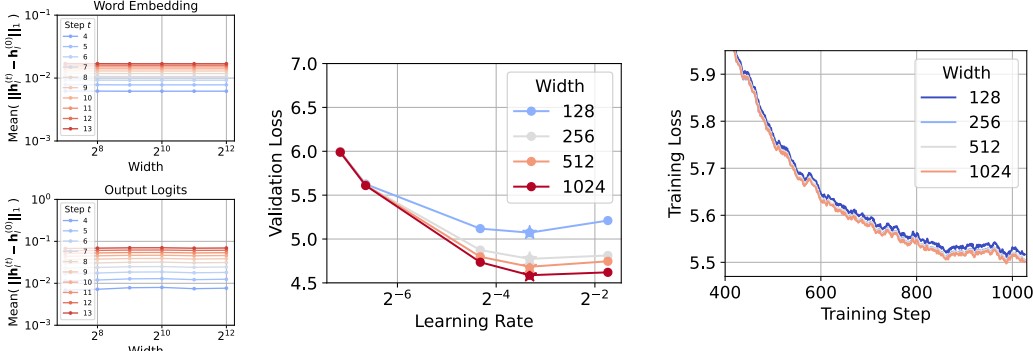

Figure 5: $\mu$P for Muon (trained on Llama2) - Coordinate check plots for the word embedding and output logits layers (left); Zero-shot learning rate transfer across increasing model width (middle); Decreasing training loss with increasing model width (right).

## B  Simulations

Consistent with existing literature, we first verify $\mu$P for ADOPT, Sophia, LAMB and Shampoo optimizers by implementing the derived parameterization scheme (Table 2) in the NanoGPT codebase Karpathy (2022). Although prior works have already implemented $\mu$P for AdamW, we present the results again for completeness. Table 3 lists some of the settings for our experimental setup to test $\mu$P on NanoGPT. Further, we demonstrate the effectiveness for AdamW, ADOPT, LAMB and Sophia on the Llama2 model, the experimental setup for which is listed in Table 15.

We also present simulation results for depth-scaling parameterization for the above optimizers on NanoGPT, using the implementation suggested in Yang et al. (2023b) and dey2025don. Note that deriving proper depth-scaling parameterization for different optimizers is an ongoing work, and we only present preliminary results on the NanoGPT codebase in Section B.2 to motivate further theoretical analysis. Table 4 lists some of the settings for our experimental setup to test the depth-scaling parameterization.

The remainder of this section documents the simulation results for AdamW (Subsection B.2.1 and Subsection B.3.1), ADOPT (Subsection B.2.2 and Subsection B.3.2), Sophia (Subsection B.2.3 and Subsection B.3.4), LAMB (Subsection B.2.4 and Subsection B.3.3) and Shampoo (Subsection B.2.5) optimizers. For each optimizer we first present the coordinate check plots under standard parameterization, $\mu$P and depth-scaling parameterization. These plots serve as a quick implementation check to monitor whether the weights blow-up, diminish to zero or remain stable with increasing model size (see discussion in (Yang et al., 2021, Section D.1, pg. 27)). We then provide tables and plots listing the validation loss for different learning rates, and increasing model width and model depth. The values in the tables for NanoGPT are the average loss values observed over multiple runs. While we do not document the standard deviations in the tables, they are highlighted in the plots. Note that since we are using an early stopping criterion for simulations performed on NanoGPT, we rely more on the observations gained from the validation loss data than the training loss data. Similar validation loss tables are documented for simulations performed on Llama2.

### B.1  Discussions

Overall, it is observed that the implementation of $\mu$P following Table 2 is quite stable with increasing model width. This is illustrated in the coordinate check plots for all the optimizers (Figs. 6 - 10 and Figs. 14 - 17 ). Under standard parameterization, the top row of the coordinate check plots shows that the relative mean of the feature vectors blow-up with increasing model width. With the incorporation of $\mu$P in the codebase, the relative mean values of the feature vectors stabilize with increasing model width (middle row of coordinate check plots).

It is interesting to note that since the theoretical underpinnings for $\mu$P hold in infinite width (Yang & Hu (2020)), the model width has to be "large enough" for the coordinate check plots to stabilize. This is especially observed in the coordinate check plots for LAMB (Fig. 9 and Fig. 16) where the mean values of the feature vectors initially increase, but gradually stabilize with increasing model width. This phenomenon is also observed in Fig. 2 which demonstrate the zero-shot learning rate transfer across model width on the NanoGPT model. In the minimum validation loss tables for ADOPT (Table 7) and LAMB (Table 11) the optimal value of the learning rate gradually stabilizes after a width of 256, whereas for AdamW (Table 5) and Sophia (Table 9) the optimal learning rate stabilizes after a width of 128. These inconsistencies across optimizers also suggest that introducing a "base model width" for $\mu$P scalings will introduce another HP. Therefore, we fix the value of the base model width to 1 in our implementation. In comparison to NanoGPT, the width scaling plots (Fig. 3) for Llama2 show that the model is "large enough" for the optimal learning rate to stabilize from the smallest model width of 128. This is perhaps because for width of 128, the total number of parameters in Llama2 is significantly higher than the total number of parameters in NanoGPT.

The second set of simulations empirically evaluate the performance of the depth-scaling parameterization in existing works (Yang et al. (2023b); Dey et al. (2025)). The coordinate check plots (bottom row) for depth-scaling demonstrate that the feature vectors are stable with increasing model depth. In the coordinate check plots for ADOPT and LAMB (Figs. 7 and 9) the feature vectors stabilize after a depth of 16, while for AdamW, Sophia and Shampoo (Figs. 6, 8 and 10) the feature vectors are stable for shallow depths too. This phenomenon is similar to our observations for $\mu$P, because

the depth-scaling parameterization is also derived for an infinite depth limit (Yang et al. (2023b)). Therefore, to prevent tuning an additional "base model depth" HP, we fix its value to 1 in our simulation setup. However, the loss plots in Figs. 11, 12 and 13 do not consistently demonstrate zero-shot learning rate transfer across increasing model depths. While the validation loss tables for AdamW (Table 6) and Sophia (Table 10) demonstrate that the optimal value of the learning rate stabilizes for deep models, the same is not observed for ADOPT (Table 8), LAMB (Table 12) and Shampoo (Table 14), where the value of the optimal learning rate oscillates as the depth is increased. These results suggest that deriving depth-scaling parameterization for different optimizers needs a more thorough theoretical analysis. Additionally, performing simulations on a finer grid of learning rates can also give further insights into the depth-scaling behavior.

## B.2   $\mu$P ON NANOGPT

Table 3: Hyperparameter values and training settings to test $\mu$P on NanoGPT model.

| Architecture | NanoGPT Karpathy (2022) |
|---|---|
| Width | 128 (scaled to 2048) |
| Depth | 8 |
| Number of heads | 2 |
| Total parameters | 1.59 M (scaled to 403 M) |
| Dataset | Tiny Shakespeare |
| Vocab size | 65 |
| Tokens per iteration | 8192 |
| Batch size | 2 |
| Stopping criteria | Early stopping if validation loss doesnot improve in last 150 iterations |
| Optimizers | AdamW / ADOPT / LAMB / Sophia / Shampoo |
| Hyperparameter search range | $\eta \in [2 \times 10^{-1}, 2 \times 10^{-5}]$ |

Table 4: Hyperparameter values and training settings to test depth-scaling parameterization on NanoGPT model.

| Architecture | NanoGPT Karpathy (2022) |
|---|---|
| Width | 256 |
| Depth | 2 (scaled to 64) |
| Total parameters | 1.6 M (scaled to 50.56 M) |
| Dataset | Tiny Shakespeare |
| Vocab size | 65 |
| Tokens per iteration | 8192 |
| Batch size | 2 |
| Stopping criteria | Early stopping if validation loss doesnot improve in last 150 iterations |
| Optimizers | AdamW / ADOPT / LAMB / Sophia / Shampoo |
| Hyperparameter search range | $\eta \in [2 \times 10^{-1}, 2 \times 10^{-5}]$ |

### B.2.1   ADAMW OPTIMIZER

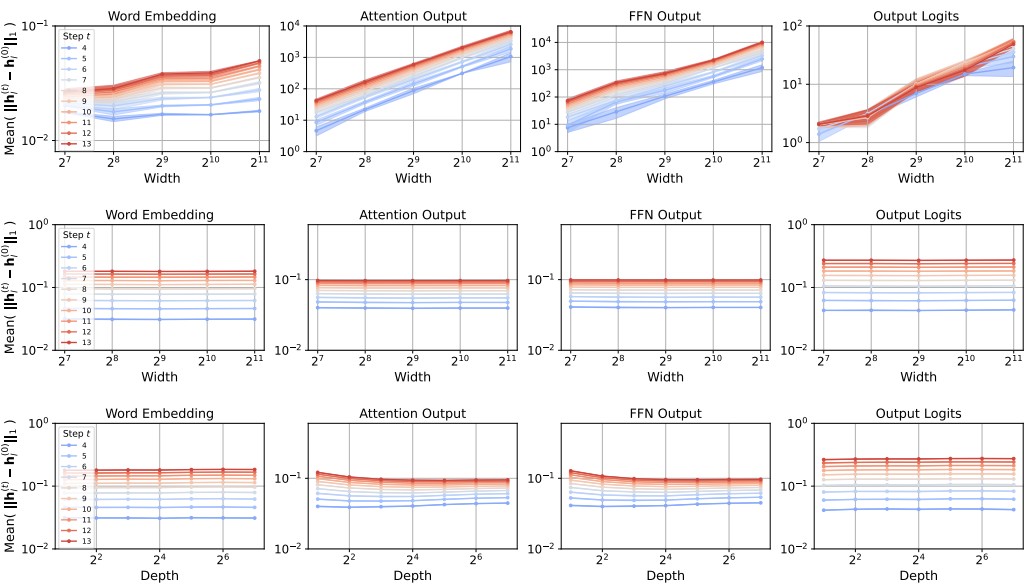

Figure 6: Coordinate check plots for AdamW under standard parameterization (top row), $\mu$P (middle row); depth scaling (bottom row) for NanoGPT model.

Table 5: Mean validation loss for increasing model width and different learning rates for AdamW on NanoGPT model. The minimum loss for each width is highlighted in green.

| LR / Width | 128 | 256 | 512 | 1024 | 2048 |
|---|---|---|---|---|---|
| $2 \times 10^{-1}$ | 2.54111195 | 2.54770319 | 2.50132585 | 2.53559383 | 2.45719266 |
| $2 \times 10^{-2}$ | 2.57009896 | 2.56583707 | 2.57900651 | 2.53385917 | 2.51431378 |
| $2 \times 10^{-3}$ | 2.63474766 | 2.6022807 | 2.64679337 | 2.63449661 | 2.55710355 |
| $2 \times 10^{-4}$ | 3.38827054 | 3.5544157 | 3.38896998 | 3.44941664 | 3.44561863 |
| $2 \times 10^{-5}$ | 4.09221347 | 4.08871428 | 4.05257797 | 4.08837303 | 4.08405908 |

Table 6: Mean validation loss for increasing model depth and different learning rates for AdamW on NanoGPT model. The minimum loss for each depth is highlighted in green.

| LR / Depth | 2 | 4 | 8 | 16 | 32 | 64 |
|---|---|---|---|---|---|---|
| $2 \times 10^{-1}$ | 2.53525917 | 2.55192765 | 2.53510944 | 2.50357556 | 2.51294963 | 2.53008548 |
| $5 \times 10^{-2}$ | 2.52700798 | 2.49422677 | 2.50334986 | 2.29428236 | 2.45176029 | 2.36860998 |
| $2 \times 10^{-2}$ | 2.55682977 | 2.52176666 | 2.56583563 | 2.30422862 | 2.45500112 | 2.5650301 |
| $2 \times 10^{-3}$ | 2.59745781 | 2.63078475 | 2.60228316 | 2.61588136 | 2.64065663 | 2.65051214 |
| $2 \times 10^{-4}$ | 3.41396125 | 3.41677833 | 3.55441554 | 3.45801504 | 3.43285489 | 3.47577778 |
| $2 \times 10^{-5}$ | 4.09297959 | 4.05970796 | 4.08871428 | 4.08113146 | 4.06712834 | 4.10902596 |

## B.2.2 ADOPT OPTIMIZER

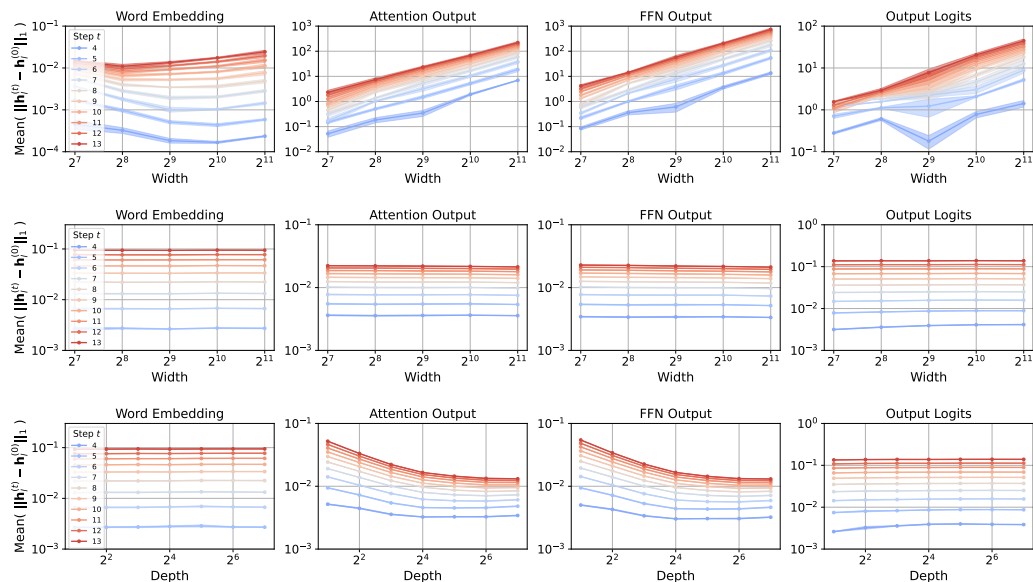

Figure 7: Coordinate check plots for ADOPT optimizer under SP (top row); $\mu$P (middle row); depth scaling (bottom row) for NanoGPT model.

Table 7: Mean validation loss for increasing model width and different learning rates for ADOPT on NanoGPT model. The minimum loss for each width is highlighted in green.

| LR / Width | 128 | 256 | 512 | 1024 | 2048 |
|---|---|---|---|---|---|
| $2 \times 10^{-1}$ | 2.55120134 | 2.54616404 | 2.54178079 | 2.5524296 | 2.54457998 |
| $7 \times 10^{-2}$ | 2.48560476 | 2.44316975 | 2.37087123 | 2.50733534 | 2.50883015 |
| $2 \times 10^{-2}$ | 2.43175697 | 2.58847451 | 2.57006375 | 2.54323697 | 2.53191725 |
| $2 \times 10^{-3}$ | 2.63016931 | 2.6073552 | 2.65681744 | 2.66118956 | 2.55337548 |
| $2 \times 10^{-4}$ | 3.528404 | 3.49065232 | 3.49065232 | 3.42789133 | 3.43255997 |
| $2 \times 10^{-5}$ | 4.09183598 | 4.08832375 | 4.0521698 | 4.08806594 | 4.08391444 |

Table 8: Mean validation loss for increasing model depth and different learning rates for ADOPT on NanoGPT model. The minimum loss for each depth is highlighted in green.

| LR / Depth | 2 | 4 | 8 | 16 | 32 | 64 |
|---|---|---|---|---|---|---|
| $2 \times 10^{-1}$ | 2.56129368 | 2.51452438 | 2.54788987 | 2.51456078 | 2.52271922 | 2.55469418 |
| $9 \times 10^{-2}$ | 2.48695572 | 2.47477563 | 2.53124801 | 2.48145302 | 2.50687472 | 2.54724765 |
| $2 \times 10^{-2}$ | 2.56718413 | 2.50419029 | 2.58847276 | 2.44447954 | 2.54996069 | 2.52524622 |
| $2 \times 10^{-3}$ | 2.67992798 | 2.62949713 | 2.6073552 | 2.60433618 | 2.61753988 | 2.6286815 |
| $2 \times 10^{-4}$ | 3.41052596 | 3.46538957 | 3.56757394 | 3.47856442 | 3.43608022 | 3.56190586 |
| $2 \times 10^{-5}$ | 4.09267759 | 4.05929391 | 4.08832375 | 4.08074443 | 4.06675259 | 4.10877307 |

## B.2.3 SOPHIA OPTIMIZER

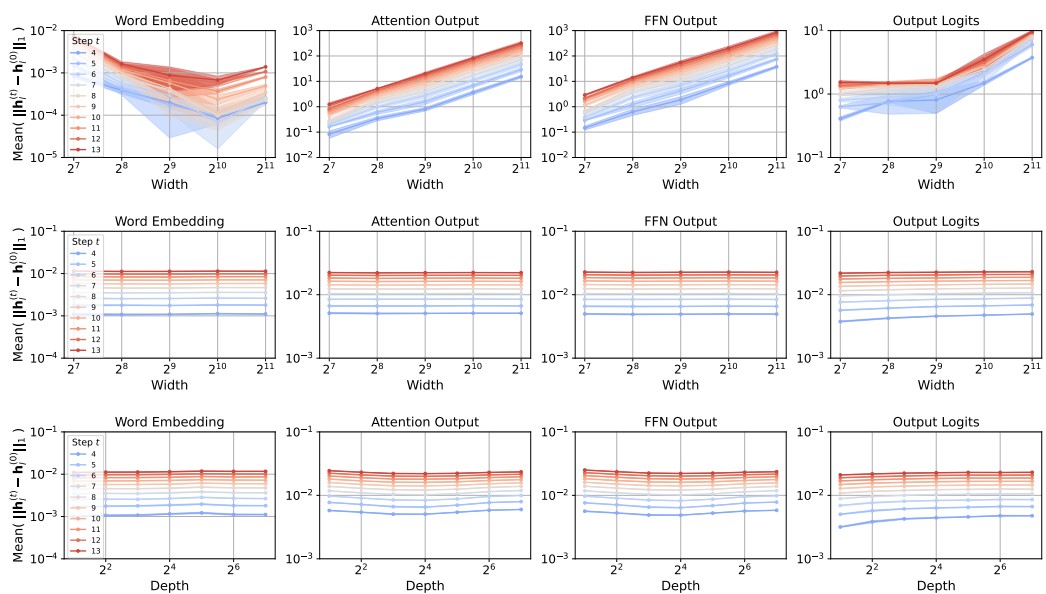

Figure 8: Coordinate check plots for Sophia optimizer under SP (top row); $\mu$P (middle row); depth scaling (bottom row) for NanoGPT model.

Table 9: Mean validation loss for increasing model width and different learning rates for Sophia on NanoGPT model. The minimum loss for each width is highlighted in green.

| LR / Width | 128 | 256 | 512 | 1024 | 2048 |
|---|---|---|---|---|---|
| $2 \times 10^{-1}$ | 3.0969398 | 2.57144117 | 2.56875261 | 2.62573036 | 2.57240287 |
| $2 \times 10^{-2}$ | 2.27450609 | 2.27830847 | 2.31632638 | 2.53347905 | 1.98427689 |
| $2 \times 10^{-3}$ | 2.5456597 | 2.61430057 | 2.5594302 | 2.54869485 | 2.65462987 |
| $2 \times 10^{-4}$ | 3.35409013 | 3.54614369 | 3.36089802 | 3.35862382 | 3.36431138 |
| $2 \times 10^{-5}$ | 4.08766381 | 4.08859126 | 4.06069756 | 4.08811712 | 4.08371623 |

Table 10: Mean validation loss for increasing model depth and different learning rates for Sophia on NanoGPT model. The minimum loss for each depth is highlighted in green.

| LR / Depth | 2 | 4 | 8 | 16 | 32 | 64 |
|---|---|---|---|---|---|---|
| $2 \times 10^{-1}$ | 2.5213503 | 3.01081316 | 3.22649105 | 3.34855215 | 3.24310446 | 3.12229093 |
| $2 \times 10^{-2}$ | 2.4717048 | 2.27232289 | 2.24736114 | 2.47475751 | 2.46061246 | 1.93401444 |
| $2 \times 10^{-3}$ | 2.54103192 | 2.58136233 | 2.61035593 | 2.610612 | 2.45068415 | 2.55488427 |
| $2 \times 10^{-4}$ | 3.40887721 | 3.52765425 | 3.54587563 | 3.40669481 | 3.33997742 | 3.47574107 |
| $2 \times 10^{-5}$ | 4.09267314 | 4.06576761 | 4.08859126 | 4.08140405 | 4.066552 | 4.10874732 |

### B.2.4 LAMB OPTIMIZER

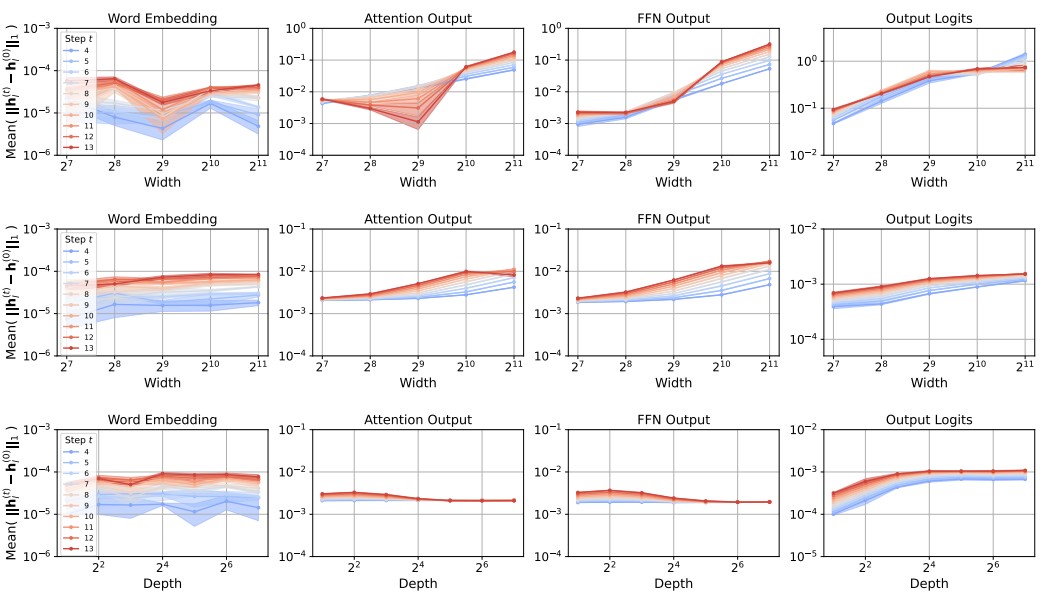

Figure 9: Coordinate check plots for LAMB optimizer under SP (top row); $\mu$P (middle row); depth scaling (bottom row) for NanoGPT model.

Table 11: Mean validation loss for increasing model width and different learning rates for LAMB on NanoGPT model. The minimum loss for each width is highlighted in green.

| LR / Width | 128 | 256 | 512 | 1024 | 2048 |
|---|---|---|---|---|---|
| $2 \times 10^{-1}$ | 3.3306915 | 2.91992474 | 2.75658234 | 2.84724092 | 2.84511503 |
| $2 \times 10^{-2}$ | 2.27427769 | 2.55330944 | 2.53250345 | 2.50694895 | 2.51612274 |
| $2 \times 10^{-3}$ | 2.46762419 | 2.42723028 | 2.47571055 | 2.49152549 | 2.46575729 |
| $2 \times 10^{-4}$ | 3.69672974 | 3.70961714 | 3.66877778 | 3.2370429 | 3.37923479 |
| $2 \times 10^{-5}$ | 4.16929531 | 4.1694754 | 4.1684103 | 4.1674579 | 4.16771809 |

Table 12: Mean validation loss for increasing model depth and different learning rates for LAMB on NanoGPT model. The minimum loss for each depth is highlighted in green.

| LR / Depth | 2 | 4 | 8 | 16 | 32 | 64 |
|---|---|---|---|---|---|---|
| $2 \times 10^{-1}$ | 2.76534136 | 2.85949779 | 2.88115621 | 3.26932732 | 3.24093787 | 3.097018 |
| $2 \times 10^{-2}$ | 2.50858307 | 2.51164389 | 2.55355501 | 2.33967662 | 2.48308444 | 2.11406271 |
| $7 \times 10^{-3}$ | 2.45117172 | 2.46691815 | 2.50231234 | 2.45691435 | 2.48629936 | 2.45780365 |
| $2 \times 10^{-3}$ | 2.50483624 | 2.54284684 | 2.42723123 | 2.43291903 | 2.43262172 | 2.42000318 |
| $2 \times 10^{-4}$ | 3.6441706 | 3.79367606 | 3.70963343 | 3.57373738 | 3.61402575 | 3.42223287 |
| $2 \times 10^{-5}$ | 4.16981506 | 4.1691486 | 4.1694754 | 4.16932933 | 4.16817395 | 4.16773876 |

### B.2.5 SHAMPOO OPTIMIZER

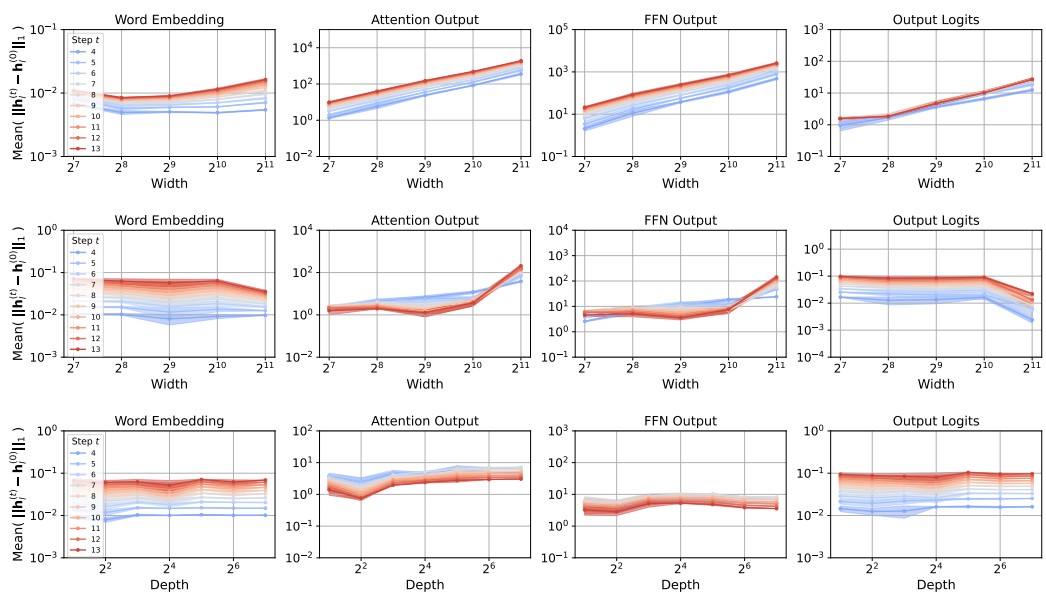

Figure 10: Coordinate check plots for Shampoo optimizer under SP (top row); $\mu$P (middle row); depth scaling (bottom row) for NanoGPT model.

Table 13: Mean validation loss for increasing model width and different learning rates for Shampoo on NanoGPT model. The minimum loss for each width is highlighted in green.

| LR / Width | 128 | 256 | 512 | 1024 | 2048 |
|---|---|---|---|---|---|
| $1 \times 10^{-2}$ | 2.64432065 | 3.00841006 | 3.26729711 | 3.39512682 | 4.17380921 |
| $9 \times 10^{-3}$ | 2.6650331 | 2.89549454 | 3.20741065 | 3.45321918 | 3.41602135 |
| $5 \times 10^{-3}$ | 2.63122805 | 2.67693043 | 3.30215279 | 3.32265353 | 3.36052688 |
| $3 \times 10^{-3}$ | 2.67303157 | 2.85103401 | 3.37194387 | 3.46975843 | 3.49201838 |
| $1 \times 10^{-3}$ | 2.90583165 | 2.97975628 | 3.61035117 | 3.57224735 | 3.72281067 |

Table 14: Mean validation loss for increasing model depth and different learning rates for Shampoo on NanoGPT model. The minimum loss for each depth is highlighted in green.

| LR / Depth | 2 | 4 | 8 | 16 | 32 | 64 |
|---|---|---|---|---|---|---|
| $3 \times 10^{-2}$ | 2.83468819 | 2.94637481 | 3.3811605 | 3.27378623 | 3.32534583 | 3.31375853 |
| $1 \times 10^{-2}$ | 2.63917089 | 2.6383814 | 2.66823014 | 3.2278808 | 3.24864435 | 3.20088768 |
| $7 \times 10^{-3}$ | 2.64190022 | 2.61007253 | 2.73991227 | 3.12863938 | 3.20985778 | 3.37485345 |
| $5 \times 10^{-3}$ | 2.77703945 | 2.72295157 | 2.72794461 | 2.93629122 | 3.25431808 | 3.37258538 |
| $3 \times 10^{-3}$ | 2.7143542 | 2.97368789 | 2.85365486 | 3.32030662 | 3.27988537 | 3.40830247 |

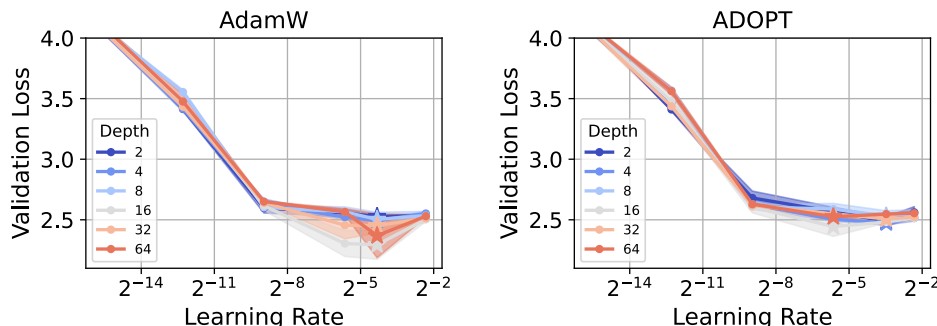

Figure 11: Mean validation loss for increasing model depth and different learning rates for AdamW (left) and ADOPT (right) on NanoGPT model.

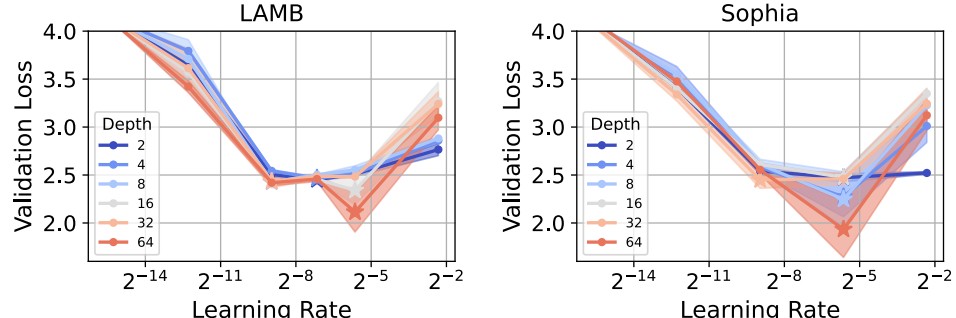

Figure 12: Mean validation loss for increasing model depth and different learning rates for LAMB (left) and Sophia (right) on NanoGPT model.

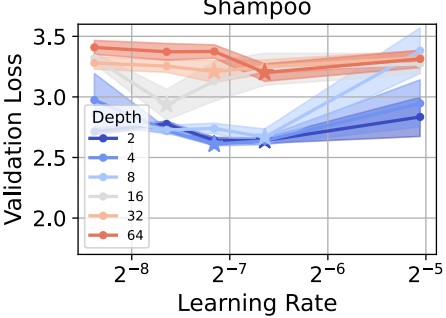

Figure 13: Mean validation loss for increasing model depth and different learning rates for Shampoo on NanoGPT model.

### B.3 $\mu$P ON LLAMA2

Table 15: Hyperparameter values and training settings to test $\mu$P on Llama2 model.

| Architecture | Llama 2 |
|---|---|
| Width | 256 (scaled to 2048) |
| Depth | 16 |
| Number of attention heads | 32 |
| Total parameters | 154M (scaled to 1.38 B) |
| Dataset | Wikitext-103 |
| Sequence length | 4096 |
| Vocab size | 32000 |
| Training set tokens | 100M |
| Batch size | 192 |
| Training steps | 1026 |
| LR decay style | cosine rule, 51 steps warm-up |
| Optimizer | AdamW / ADOPT / LAMB / Sophia |
| Weight decay | 0.1 |
| Dropout | 0.0 |
| $\mu$P HP search range | $\eta \in [5 \times 10^{-1}, 5 \times 10^{-4}]$ |

#### B.3.1 ADAMW

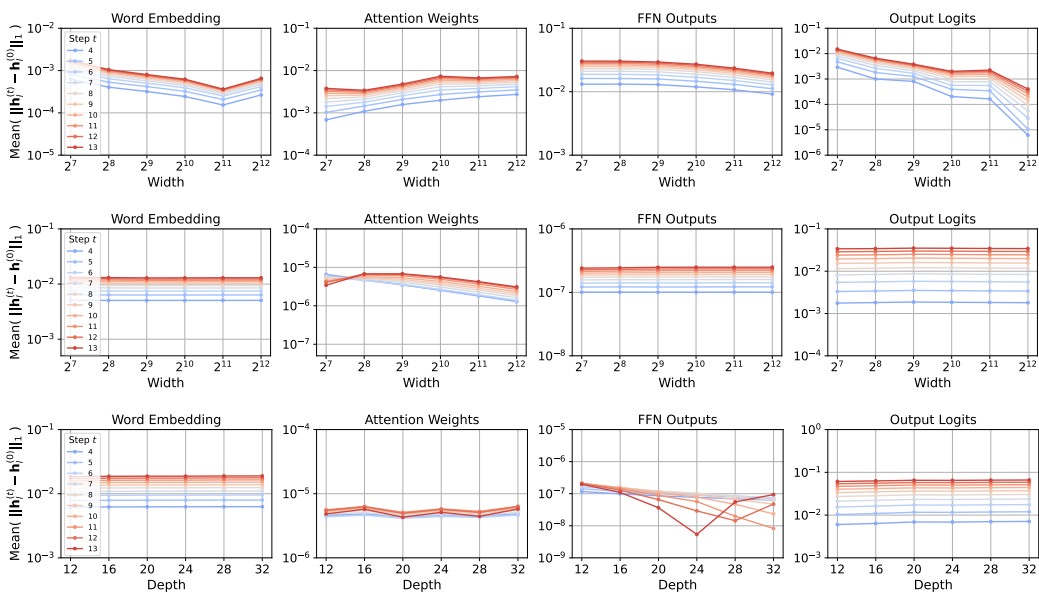

Figure 14: Coordinate check plots for AdamW optimizer under SP (top row); $\mu$P (middle row); depth scaling (bottom row) for Llama2 model.

Table 16: Validation loss for increasing model width and different learning rates for AdamW on Llama2 model. The minimum loss for each width is highlighted in green.

| LR / Width | 128 | 256 | 512 | 1024 | 2048 |
|---|---|---|---|---|---|
| $5 \times 10^{-1}$ | 4.55491 | 4.02676 | 3.81251 | 3.73573 | 3.79477 |
| $3 \times 10^{-1}$ | 4.24978 | 3.90242 | 3.83252 | 3.89484 | 3.75046 |
| $1 \times 10^{-1}$ | 4.48696 | 4.21314 | 4.05265 | 4.02101 | 3.95419 |
| $5 \times 10^{-2}$ | 4.70421 | 4.4353 | 4.39753 | 4.34169 | 4.31635 |
| $1 \times 10^{-1}$ | 5.57795 | 5.56284 | 5.56173 | 5.55771 | 5.55774 |

### B.3.2 ADOPT

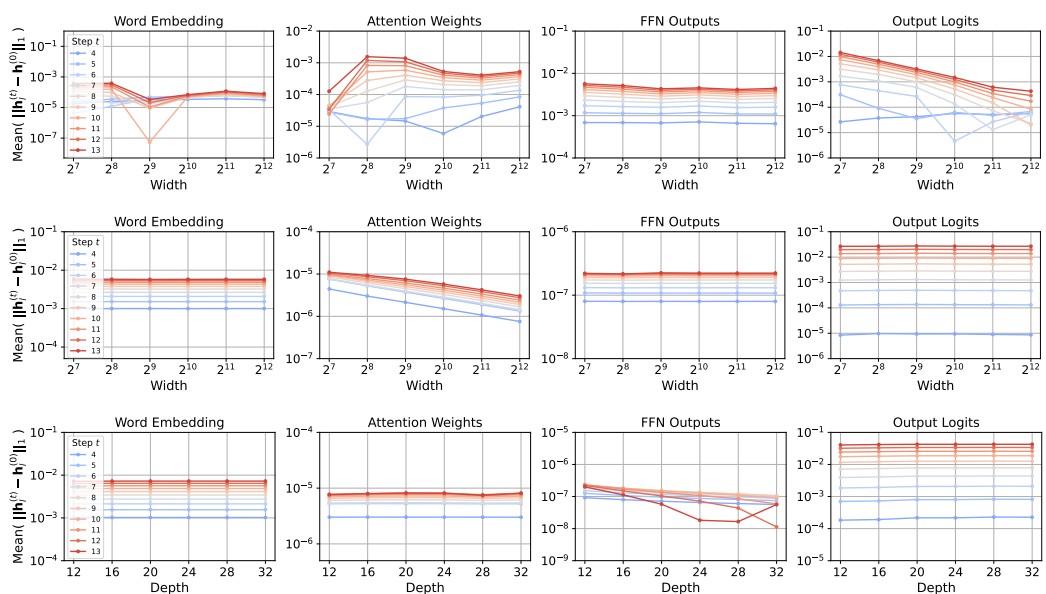

Figure 15: Coordinate check plots for ADOPT optimizer under SP (top row); $\mu$P (middle row); depth scaling (bottom row) for Llama2 model.

Table 17: Validation loss for increasing model width and different learning rates for ADOPT on Llama2 model. The minimum loss for each width is highlighted in green.

| LR / Width | 128 | 256 | 512 | 1024 | 2048 |
|---|---|---|---|---|---|
| $5 \times 10^{-1}$ | 4.39033 | 4.02007 | 3.83932 | 3.77732 | 3.76814 |
| $3 \times 10^{-1}$ | 4.11789 | 3.85536 | 3.72552 | 3.67802 | 3.66973 |
| $2 \times 10^{-1}$ | 4.23765 | 3.87949 | 3.78242 | 3.80016 | 3.78846 |
| $1 \times 10^{-1}$ | 4.32335 | 4.07597 | 3.9912 | 3.91654 | 3.95519 |
| $7 \times 10^{-2}$ | 4.43819 | 4.22574 | 4.13565 | 4.06852 | 4.0683 |
| $5 \times 10^{-2}$ | 4.64121 | 4.38096 | 4.31582 | 4.22186 | 4.21248 |

### B.3.3 LAMB

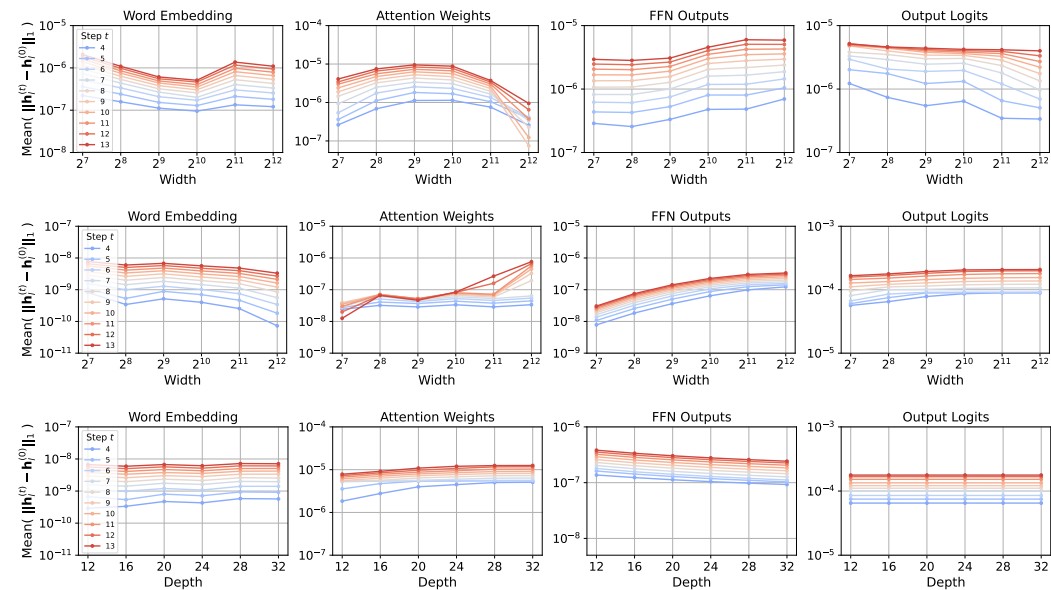

Figure 16: Coordinate check plots for LAMB optimizer under SP (top row); $\mu$P (middle row); depth scaling (bottom row) for Llama2 model.

Table 18: Validation loss for increasing model width and different learning rates for LAMB on Llama2 model. The minimum loss for each width is highlighted in green.

| LR / Width | 128 | 256 | 512 | 1024 | 2048 |
|---|---|---|---|---|---|
| $3 \times 10^{-2}$ | 7.18452 | 6.35059 | 6.0384 | 6.52966 | 6.13429 |
| $1 \times 10^{-2}$ | 5.58878 | 5.5638 | 5.56049 | 5.79174 | 6.01439 |
| $5 \times 10^{-3}$ | 6.57476 | 6.60454 | 6.66398 | 6.98093 | 7.0471 |
| $1 \times 10^{-3}$ | 10.25112 | 10.23998 | 10.22575 | 10.21199 | 10.19599 |
| $5 \times 10^{-4}$ | 10.32997 | 10.32776 | 10.32398 | 10.32062 | 10.31677 |

### B.3.4 SOPHIA

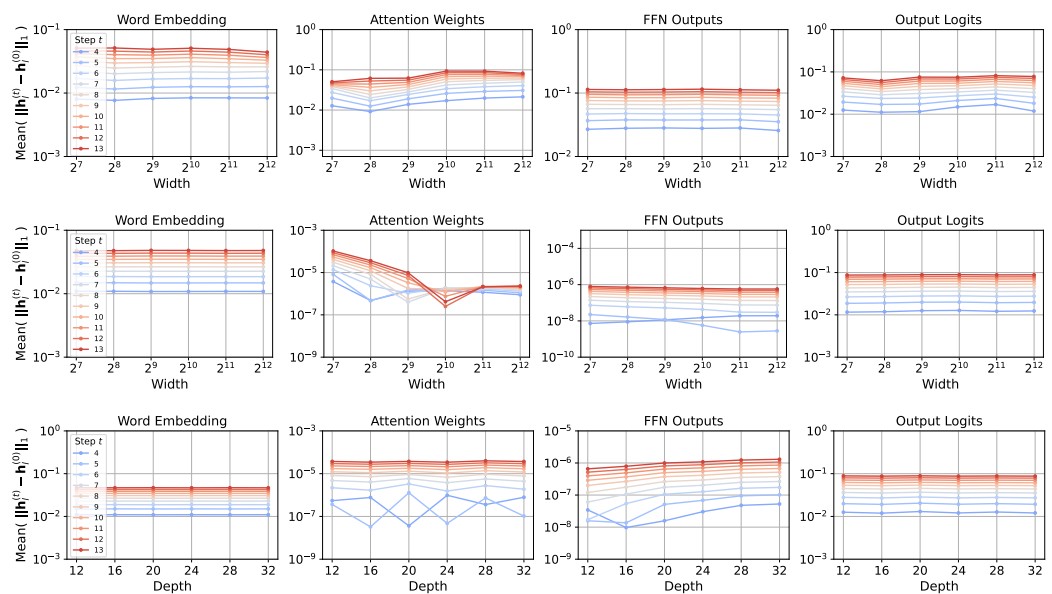

Figure 17: Coordinate check plots for Sophia optimizer under SP (top row); $\mu$P (middle row); depth scaling (bottom row) for Llama2 model.

Table 19: Validation loss for increasing model width and different learning rates for Sophia on Llama2 model. The minimum loss for each width is highlighted in green.

| LR / Width | 128 | 256 | 512 | 1024 | 2048 |
|---|---|---|---|---|---|
| $5 \times 10^{-1}$ | 7.19403 | 6.99576 | 6.68992 | 6.60376 | 6.31375 |
| $3 \times 10^{-1}$ | 6.17604 | 5.90826 | 5.80694 | 5.6738 | 5.71962 |
| $1 \times 10^{-1}$ | 4.14122 | 3.83654 | 3.75926 | 3.67419 | 3.62891 |
| $7 \times 10^{-2}$ | 4.42758 | 4.31702 | 4.05756 | 3.93561 | 3.94189 |
| $5 \times 10^{-2}$ | 4.76632 | 4.51022 | 4.41358 | 4.34452 | 4.30914 |
| $3 \times 10^{-2}$ | 4.82305 | 4.79592 | 4.73067 | 4.67473 | 4.74689 |

