# OpenReview forum: "Extending $\mu$P: Spectral Conditions for Feature Learning Across Optimizers"
_ICLR.cc/2026/Conference — Submitted to ICLR 2026_

### Official Review · Reviewer_qLDZ · 2025-10-26

**Soundness:** 3
**Presentation:** 3
**Contribution:** 2
**Rating:** 4
**Confidence:** 3

**Summary:**

The authors derive $\mu$P for a wide range of standard optimizers (AdamW, LAMB, Sophia, Shampoo) via spectral scaling and demonstrate zero-shot hyperparameter transfer across scales and benchmarks.

**Strengths:**

1. The derivations presented in the paper are nontrivial, detailed, and well-written.
2. The proposed method is indeed easier to work with than previous formulations of $\mu$P.
3. The empirical results support the theoretical claims and are promising.

**Weaknesses:**

1. My main concern with the results is the scalability of proposed method. Given that the paper is primarily empirical in nature, I would find the results a lot more convincing if the experimental setup involved larger (i.e. 10B+) models.
2. The derivations provided rely on simplifying assumptions, such as $\beta_1=\beta_2=\epsilon=0$, which are atypical regimes. I suggest to include either an analysis with momentum and weight decay considered or a discussion of why the simplified regime is sufficient.
3. Several assumptions are hidden away in the appendix, with little discussion provided on their practicality.
4. Muon is mentioned as a state-of-the-art optimizer, but a $\mu$P derivation for Muon is not provided.
5. Minor nitpicks: use `\citep{}` instead of `(\cite{})` for parenthetical citations. The overflow on page 9 is visually unappealing.

**Questions:**

See Weaknesses

---

> ### Author Response · Authors · 2025-11-26
> **Scalability of proposed method**
>
> We thank the reviewer for their detailed feedback and constructive criticism. We have incorporated as many suggestions as possible in the revised manuscript and we are looking forward to further discussion.
>
> $\text{Ans}1.$
>
> Perhaps it was not clear in the original draft of our manuscript, but the primary contribution of this work is the analytical derivation of $\mu$P scaling laws for different optimizers. The empirical results are a separate contribution, which do not aid in the derivations, rather demonstrate the optimal learning rate transfer and wider is better phenomenon based on our derivations (Table 2) for different optimizers. Therefore, the scaling laws hold for models of any width.
>
> We realize that this distinction was not clearly made in the original draft of the manuscript, so we have amended the introduction section to highlight that the derivations and the implementations are two separate contributions of this work. Implementing $\mu$P in large codebases and for several different optimizers is not a trivial task, and the challenges of incorporating $\mu$P into code are highlighted in previous works [1,4]. One of the key challenges in moving from theory to implementation is identifying the set of hyperparameters which should also be scaled to obtain the desired performance. For example, while initial works suggested that for small values of $\epsilon$, the hyperparameter doesn't have to be scaled [5], more recent empirical studies suggest that in low precision training regimes, scaling $\epsilon$ improves overall performance and it is now widely practiced in implementation [1,2]. In this work, we not only provide a rigorous way of deriving scaling laws for relevant hyperparameters (see section 4.2), but we also identify and implement these scaling laws for the hyperparameters corresponding to different optimizers to obtain expected performance. In other words, the empirical results validate our implementation rather than validating the derived scaling laws. The code for both NanoGPT and Llama2 will be made public so other users can directly start working from our $\mu$P implementations.
>
> Finally, our choice of experimental setup and model sizes are guided by existing works [1,2,3] which use comparable models to demonstrate $\mu$P. However, in contrast to existing works that typically implement and study $\mu$P for a single optimizer, we have implemented $\mu$P for six different optimizers on two different LLMs. A study on a 10+B parameter model for all the optimizers will require huge amount of computing resources which is currently outside the scope of this work but is part of a future work which is underway.
>
> REFERENCES:
>
> [1] Charlie Blake, Constantin Eichenberg, Josef Dean, Lukas Balles, Luke Yuri Prince, Bj¨orn Deiseroth, Andres Felipe Cruz-Salinas, Carlo Luschi, Samuel Weinbach, and Douglas Orr. u-\μ p: The unit-scaled maximal update parametrization. In The Thirteenth International Conference on Learning Representations, 2025.
>
> [2] Nolan Dey, Bin Claire Zhang, Lorenzo Noci, Mufan Li, Blake Bordelon, Shane Bergsma, Cengiz Pehlevan, Boris Hanin, and Joel Hestness. Don’t be lazy: Completep enables compute-efficient deep transformers. arXiv preprint arXiv:2505.01618, 2025. URL https://doi.org/10.48550/arXiv.2505.01618.
>
> [3] Vineet Gupta, Tomer Koren, and Yoram Singer. Shampoo: Preconditioned stochastic tensor optimization. In International Conference on Machine Learning, pp. 1842–1850. PMLR, 2018. URL https://doi.org/10.48550/arXiv.1802.09568.
>
> [4] Lucas Lingle. An empirical study of μp learning rate transfer, 2025. URL: https://arxiv.org/abs/2404.05728.
>
> [5] Greg Yang, Edward Hu, Igor Babuschkin, Szymon Sidor, Xiaodong Liu, David Farhi, Nick Ryder,
> Jakub Pachocki, Weizhu Chen, and Jianfeng Gao. Tuning large neural networks via zero-shot hyper-
> parameter transfer. Advances in Neural Information Processing Systems, 34:17084–17097, 2021. URL
> https://doi.org/10.48550/arXiv.2203.03466.

---

> ### Author Response · Authors · 2025-11-26
> **Hyperparameter scalings**
>
> $\text{Ans}2.$
>
> We thank the reviewer for their thoughtful clarification regarding the hyperparameter scalings. The simplifying assumptions are only used to derive the initial scaling law for the learning rate. However, towards the end of Section 4.2, we derive the scaling laws for hyperparameters. In particular, the hyperparameters $\beta_1$ and $\beta_2$ do not need scaling because they are width independent constants and thus, their order of magnitude is $\Theta(1)$. Therefore, these terms do not contribute to the order of magnitude calculations of the momentum terms and do not have an effect on the final scaling laws. In general, the momentum terms in most optimizers are bounded and are width-independent, thereby avoiding the need to scale them.
>
> On the other hand, hyperparameters like the weight decay parameter, $\lambda$, and $\epsilon$ for numerical stability, require scaling, as has been reported by multiple existing works [1,2,3]. We provide a detailed derivation of the scaling laws for both these hyperparameters in the revised draft of the manuscript, towards the end of Section 4.2. These scaling laws have been incorporated in our code to obtain the empirical results. Further, we have included brief comments discussing the scaling of hyperparameters for all optimizers towards the end of their respective sections. It is also interesting to note that we do not need to fix $\beta_1 = \beta_2 = \epsilon = 0$ for the derivation of LAMB, because its derivation invokes some results from AdamW's analysis, thereby avoiding the need to explicitly handle the hyperparameters again in LAMB's derivation. We have highlighted the relationship between LAMB and AdamW in the revised manuscript (see Section 4.3). Similarly, the derivation of Muon (Appendix A) does not require us to handle the hyperparameters explicitly because the update matrix for Muon is orthogonal, and we know that the spectral norm of an orthogonal matrix is $1$. Based on these examples, we would like to emphasize that our proposed framework can also accommodate optimizer-specific insights into the derivations and eliminate the need to set hyperparameters to $0$ during the calculations. However, for AdamW, Sophia and Shampoo, no such insights were available and our derivations demonstrate how to handle the hyperparameters from the ground up.
>
> EFERENCES:
>
> [1] Charlie Blake, Constantin Eichenberg, Josef Dean, Lukas Balles, Luke Yuri Prince, Bj¨orn Deiseroth, Andres Felipe Cruz-Salinas, Carlo Luschi, Samuel Weinbach, and Douglas Orr. u-\μ p: The unit-scaled maximal update parametrization. In The Thirteenth International Conference on Learning Representations, 2025.
>
> [2] Nolan Dey, Bin Claire Zhang, Lorenzo Noci, Mufan Li, Blake Bordelon, Shane Bergsma, Cengiz Pehlevan, Boris Hanin, and Joel Hestness. Don’t be lazy: Completep enables compute-efficient deep transformers. arXiv preprint arXiv:2505.01618, 2025. URL https://doi.org/10.48550/arXiv.2505.01618.
>
> [3] Greg Yang, Edward Hu, Igor Babuschkin, Szymon Sidor, Xiaodong Liu, David Farhi, Nick Ryder, Jakub Pachocki, Weizhu Chen, and Jianfeng Gao. Tuning large neural networks via zero-shot hyper- parameter transfer. Advances in Neural Information Processing Systems, 34:17084–17097, 2021. URL https://doi.org/10.48550/arXiv.2203.03466.

---

> ### Author Response · Authors · 2025-11-26
> **Assumptions to move from theory to practice and $\mu$P for Muon**
>
> $\text{Ans}3.$
>
> We realize that the way assumptions were presented in the first draft of the manuscript were confusing. Therefore, we have moved the assumptions from the appendix to Section 3.3 in the revised document and added discussion for why the assumptions hold in practice. To be clear, all our derivations hold exactly for a linear MLP trained with batch size 1. The assumptions in Section 3.3 are required to justify why the derived scalings in Table 2 can be directly implemented for more complex and practically used models, including LLMs. The assumptions in Section 3.3 are easily realized in implementation because they are made on the norms of the relevant quantities and don't restrict the nature of activation functions, vectors, updates etc. significantly.
>
> We would also like to emphasize that there are always gaps between theory and practice. The assumptions on the model presented in Section 3 seem restrictive, but they allow our derivations to hold precisely and prevent significant approximation errors in the theoretical analysis  itself. In fact, the supporting results presented in Remark 1 and Remark 2 hold exactly because of the rank 1 nature of the update matrix, and these results are essential for the $\mu$P derivation for Shampoo. Note that the scalings derived for Shampoo in Section 4.5 match the scalings derived from the tensor programming approach [1] because working with such a model is standard practice in $\mu$P literature so far. By eliminating large approximation errors in the analytical derivations, the assumptions in Section 3.3 solely focus on addressing how the derived $\mu$P scalings can be applied to more complex models used in practice. This systematic delineation will also aid future studies to relax some of the assumptions made in this work.
>
> $\text{Ans}4.$
>
> We have included the derivation of $\mu$P for Muon in Appendix A. Muon is one of the first optimizers to explicitly incorporate width-independent scaling in its update rule by multiplying the learning rate with a factor of $\sqrt{\frac{n_{l}}{n_{l-1}}}$ [2]. Therefore, intuitively, Muon should not require any further scaling of the learning rate. Our derivation presented in Appendix A confirms this. Note that we are working with the most recent version of Muon [2] which is different than the version implemented in the Python library. However, we also make comments about the equivalence between Muon and Shampoo, and if a user wants to work with initial implementations of Muon (which do not multiply the learning rate with the scaling factor), then Muon follows the same $\mu$P scaling as Shampoo.
>
> $\text{Ans}5.$
>
> Thank you for pointing out the formatting issues. We have fixed the citations as well as the overflow in the derivation of Shampoo.
>
> REFERENCES:
>
> [1] Satoki Ishikawa and Ryo Karakida. On the parameterization of second-order optimization effective towards the infinite width. In The Twelfth International Conference on Learning Representations.
>
> [2] Jeremy Bernstein. Deriving muon. https://jeremybernste.in/writing/deriving-muon, 2025.

---

### Official Review · Reviewer_gp6p · 2025-10-26

**Soundness:** 2
**Presentation:** 2
**Contribution:** 2
**Rating:** 4
**Confidence:** 4

**Summary:**

This paper addresses the challenge of hyperparameter (HP) tuning for large-scale models. Maximal update parameterization ($\mu$P) is a promising technique that enables zero-shot HP transfer from small to large models, but its derivation via tensor programs is complex and has limited its application beyond SGD and Adam.

Building on recent work that proposed spectral conditions as a simpler, alternative foundation for $\mu$P, this paper proposes a generic framework that uses this spectral approach. The authors apply this framework to derive, for the first time, $\mu$P scaling rules for a wider range of popular optimizers, including AdamW, ADOPT, LAMB, Sophia, and Shampoo.

The paper validates these new derivations empirically on NanoGPT and Llama2 models, demonstrating that the new $\mu$P scaling rules successfully achieve zero-shot learning rate transfer as model width increases.

**Strengths:**

1. Addresses a High-Impact Problem: The computational cost of HP tuning is a significant bottleneck in training large models. $\mu$P is a powerful tool to mitigate this, but its applicability has been limited. Extending $\mu$P to a wider, more modern set of optimizers like LAMB, Sophia, and Shampoo is a valuable and practical contribution to the field.
2. Clear Practical Takeaways: The paper delivers actionable scaling rules for several optimizers (summarized in Table 2).
3. Strong Empirical Validation: The claims are well-supported by experiments. The plots in Figure 2, Figure 3, and Figure 4 (right) clearly demonstrate the two key benefits of $\mu$P: stable zero-shot learning rate transfer across widths and the wider is better phenomenon (decreasing training loss with width). The validation on both NanoGPT and Llama2 models strengthens the empirical claims.

**Weaknesses:**

1. Incremental Novelty: The core conceptual leap, replacing complex tensor programs with more tractable spectral conditions, was introduced by prior work. This paper's main contribution is the application of this existing spectral framework to new optimizers. While this is a useful engineering and analytical contribution, the intellectual novelty of the methodology itself is limited.
2. Repetitive and Padded Structure: The main methodological idea is presented in Section 4.1 as a Generic Framework. The following subsections (4.2-4.6) are largely straightforward, repetitive applications of this single framework to different optimizers. For example, the derivations for AdamW and Sophia are nearly identical. This section feels padded and could have been substantially condensed by focusing on the unique analytical challenges (e.g., for LAMB and Shampoo) and moving the more trivial derivations to the appendix.
3. Typesetting and Readability Issues: The paper appears to be in a draft state with major formatting errors. In Section 4.6 (Page 9), the entire multi-line equation block for the Shampoo optimizer derivation runs off the page margin, which suggests a lack of careful preparation.
4. Limited Theoretical Scope: The core derivations are performed for a simple linear MLP with a batch size of 1. The extension to complex, non-linear models like Llama2 relies on a set of assumptions outlined in Appendix A. The paper does not provide a deep analysis of why these assumptions should hold for these more complex optimizers, relying instead on the empirical results to justify the leap.

**Questions:**

1. Could the authors clarify their novel methodological contribution beyond the direct application of the spectral conditions framework? Is the generic framework in 4.1 the primary contribution, or is the contribution simply the set of new scaling rules derived from it?
2. Given the significant overlap in the analysis for AdamW, ADOPT, and Sophia, would the authors consider restructuring Section 4 to avoid repetition and better highlight the unique analytical challenges posed by LAMB and Shampoo?

---

> ### Author Response · Authors · 2025-11-26
> **Clarification on contributions and assumptions required to move from theory to practice**
>
> We thank the reviewer for their valuable comments which have helped us improve the clarity and quality of the manuscript. Please see below for a detailed answer.
>
> $\text{Ans}1.$
>
> We consider both the generic framework and the derivations of the scaling laws for different optimizers as the key contributions of this work.
> The framework proposed in this work significantly reduces the cumbersome calculations involved in the tensor programming approach and makes it intuitive to reason about $\mu$P for different optimizers. Although we use results from  a previous work which shows the equivalence between $\mu$P and spectral conditions, our framework demonstrates how to interpret the spectral norm conditions to derive the scalings for both the model architecture and different optimizers, which is not elaborately discussed in the previous work. We effectively leverage the framework to derive $\mu$P for a number of first and second order optimizers. In contrast, there are rarely any existing works which have successfully extended $\mu$P to other optimizers, primarily because of the intractable tensor program calculations. A notable exception is the work by [1] where the authors derived $\mu$P for Shampoo. However, in comparison to [1], we arrive at the same $\mu$P scalings for Shampoo by using the more tractable spectral conditions, thereby significantly reducing the burden of tedious calculations under the tensor programming approach as well as providing an easy way to extend $\mu$P to other second-order methods.
> Additionally, we have also expanded Section 4.2 to discuss which hyperparameters should be scaled under $\mu$P and how to derive the scaling laws for the different hyperparameters.
>
> We also acknowledge the reviewer's comment about not clearly justifying why our derivations should hold in practice. Therefore, we have moved assumptions from the Appendix to Section 3.3 and added insights as to why the assumptions hold in practice. To be clear, all our derivations hold exactly for a linear MLP trained with batch size 1. The assumptions in Section 3.3 are required to justify why the derived scalings in Table 2 can be directly implemented for more complex and practically used models, including LLMs. The assumptions in Section 3.3 are realized in practice because they are made on the norms of the relevant quantities and don't restrict the nature of activation functions, vectors, updates etc. significantly. We would also like to emphasize that there are always gaps between theory and practice. The assumptions on the model presented in Section 3 seem restrictive, but they allow our derivations to hold precisely and prevent significant approximation errors in the theoretical analysis. In fact, the supporting results presented in Remark 1 and Remark 2 hold exactly because of the rank 1 nature of the update matrix, and these results are essential for the $\mu$P derivation for Shampoo. Note that the scalings derived for Shampoo in Section 4.5 match the scalings derived from the tensor programming approach [1] because working with such a model is standard practice in $\mu$P literature so far. By eliminating large approximation errors in the analytical derivations, the assumptions in Section 3.3 solely focus on addressing how the derived $\mu$P scalings can be applied directly to more complex models used in practice. This systematic delineation will also aid future studies to relax some of the assumptions made in this work.
>
> We would also like to highlight that implementing $\mu$P for all the different optimizers, on multiple benchmark LLMs, is also not a trivial task. Therefore, as pointed out by other reviewers, we emphasize our implementations as a separate contribution of this work in the new draft of the manuscript. The empirical results, primarily in Fig. 2 and Fig. 3, validate our implementation by exhibiting expected behavior under $\mu$P. The code for both NanoGPT and Llama2 will be made public so other users can directly start working from our $\mu$P implementations.
>
> REFERENCES
>
> [1] Satoki Ishikawa and Ryo Karakida. On the parameterization of second-order optimization effective towards the infinite width. In The Twelfth International Conference on Learning Representations.

---

> ### Author Response · Authors · 2025-11-26
> **Adding $\mu$P derivation for Muon and restructuring Section 4**
>
> Thank you for your suggestions. The ADOPT rule can indeed be obtained directly from AdamW, as the two optimizers differ only in the order of operations. For Sophia, however, we would like to emphasize that although the final scaling laws resemble those of AdamW, their derivations are not trivially equivalent. In particular, handling the non-smooth function $\operatorname{clip}(\cdot,\cdot)$ and justifying why we obtain a tight upper bound for the final scaling result based on the value of $\gamma$ are unique aspects of Sophia's analysis. These steps illustrate how similar update rules can be analyzed in future work. For these reasons, it is important to present Sophia's derivation separately from AdamW's. After deriving Sophia's scaling rules formally, we also provide an intuitive explanation for why Sophia mirrors AdamW's update rule in the special case where the Hessian terms are negative.
>
> We have incorporated the reviewer's additional suggestions in the revised manuscript. We have moved the discussion of ADOPT and added the derivation for Muon to Appendix A. Muon is one of the first optimizers to explicitly incorporate width-independent scaling in its update rule by multiplying the learning rate by $\sqrt{n_{l} / n_{l-1}}$ [1]. Intuitively, this suggests that Muon should not require any further learning rate scaling, and our derivation in Appendix A confirms this. We have also amended Section 4.3 to expand upon LAMB's derivation and highlight its relation with AdamW. Further, we have moved the assumptions that justify applying the derived $\mu$P rules to practical models to Section~3.3. After reading the reviewer's comments, we realized that discussing the practical applicability of $\mu$P earlier in the paper improves clarity.
>
> We also appreciate the reviewer's comments regarding formatting issues in the Shampoo derivation. Although we originally felt that the steps conveyed a clear logical flow, we now recognize that this may have reduced readability. We have revised the formatting of the proof in the main text and moved a more detailed version of the derivation to Appendix A.
>
> REFERENCE:
>
> [1] Jeremy Bernstein. Deriving muon. https://jeremybernste.in/writing/deriving-muon, 2025.

---

### Official Review · Reviewer_aLhn · 2025-10-30

**Soundness:** 2
**Presentation:** 2
**Contribution:** 2
**Rating:** 4
**Confidence:** 3

**Summary:**

The paper studies $\mu$P parameterization for stable feature learning and allows zero-shot learning rate (LR) transfer across model sizes (widths). In particular, the paper proposes a spectral norm-based recipe to derive $\mu$P parameterization for a range of optimizers, including AdamW, ADOPT, LAMB, Sophia, and Shampoo. Spectral conditions on weights and updates are used to replace complex tensor programs for deriving $\mu$P. Numerical experiments on NanoGPT and Llama-2 are provided to validate the derived parameterization.

**Strengths:**

1. The proposed framework largely simplifies tensor programs, and the results are clearly presented. The derivations are simple and applicable for a range of optimizers.

2. Beyond recovering the parameterization for AdamW, the paper provides new parameterizations for LAMB and Shampoo.

**Weaknesses:**

1. Muon optimizer is mentioned in the introduction, but there is no corresponding derivation for it. See Question 2.

2. The derivations rely on strong simplifications, such as batch size=1, $\beta_1=\beta_2=\epsilon=0$, and the dropping of weight decay. It is questionable whether the derived exponents remain valid when these hyperparameters are changed.

3. The results for Shampoo (Figure 2) do not show a clear zero-shot LR transfer. The losses often worsen with width across the LR grid. See Question 3.

4. Depth scaling remains preliminary.

**Questions:**

1. In Table 2, the Init. Var. and Multiplier result in the same effect as the tensor programs. Can we adopt the scalings from the tensor program for these two?

2. Muon optimizer is mentioned in the introduction and Figure 1, but there is no derivation for it. Is it considered to be the same as Shampoo?

3. In Figure 2, the validation loss of Shampoo is very high compared to other optimizers, and the best performance does not occur for the largest width. Is it the nature of Shampoo, or does it suggest that the parameterization for Shampoo is suboptimal?

---

> ### Author Response · Authors · 2025-11-26
> **$\mu$P for Muon and hyperparameter scalings**
>
> We thank the reviewer for their constructive review and suggestions. We provide detailed answers below.
>
> $\text{Ans}1.$
>
> Yes, the scalings derived from using the proposed generic framework in Section 4.1 are equivalent to the scalings derived from using tensor programs for the initial variance and model weights. While either of the two methods can be used, we emphasize the ease and simplicity of using the spectral scaling conditions over tensor programs.
>
> $\text{Ans}2.$
>
> We thank the reviewer for encouraging us to include the $\mu$P derivation for Muon. Incorporating this suggestion into the revised draft has further demonstrated the versatility of our proposed framework.
>
> We have included the derivation of $\mu$P for Muon in Appendix A. Muon is one of the first optimizers which explicitly incorporates width-independent scaling in its update rule by multiplying the learning rate with a factor of $\sqrt{\frac{n_{l}}{n_{l-1}}}$ [1]. Therefore, intuitively, Muon should not require any further scaling of the learning rate. Our derivation presented in Appendix A confirms this. Note that we are working with the most recent version of Muon [1] which is different than the version implemented in the Python library. However, we also comment on the equivalence between Muon and Shampoo, and if a user wants to work with initial implementations of Muon (which do not multiply the learning rate with the scaling factor), then Muon follows the same $\mu$P scaling as Shampoo.
>
> In addition to Muon's analysis, we have also addressed the reviewer's concern about deriving $\mu$P by setting some of the hyperparameters to 0. We have expanded our discussion on deriving the scalings of hyperparameters towards the end of Section 4.2. In particular, momentum terms $\beta_1$ and $\beta_2$ do not require any additional scaling because they have $\Theta(1)$ order of magnitude and do not change the analysis of the momentum terms, whether they are explicitly included in the derivation or not. However, we include the scalings for the weight decay parameter $\lambda$ and $\epsilon$, which affect the numerical stability of $\mu$P [2,3]. We have also included a brief discussion on scaling the hyperparameters for all optimizers towards the end of their respective sections. It is also interesting to note that we do not need to fix $\beta_1 = \beta_2 = \epsilon = 0$ for the derivation of LAMB, because its derivation invokes some results from AdamW's analysis, thereby avoiding the need to explicitly handle the hyperparameters again in LAMB's derivation. We have highlighted the relationship between LAMB and AdamW in the revised manuscript (see Section 4.3). Similarly, the derivation of Muon (Appendix A) does not require us to handle the hyperparameters explicitly because the update matrix for Muon is orthogonal, and we know that the spectral norm of an orthogonal matrix is $1$. Based on these examples, we would like to emphasize that our proposed framework can also accommodate optimizer-specific insights into the derivations and eliminate the need to set hyperparameters to $0$ during the calculations. However, for AdamW, Sophia and Shampoo, no such insights are available and instead, our derivations demonstrate how to handle the hyperparameters from the ground up.
>
> REFERENCES:
>
> [1] Jeremy Bernstein. Deriving muon. https://jeremybernste.in/writing/deriving-muon, 2025.
>
> [2] Charlie Blake, Constantin Eichenberg, Josef Dean, Lukas Balles, Luke Yuri Prince, Bj¨orn Deiseroth, Andres Felipe Cruz-Salinas, Carlo Luschi, Samuel Weinbach, and Douglas Orr. u-\μ p: The unit-scaled maximal update parametrization. In The Thirteenth International Conference on Learning Representations, 2025.
>
> [3] Nolan Dey, Bin Claire Zhang, Lorenzo Noci, Mufan Li, Blake Bordelon, Shane Bergsma, Cengiz Pehlevan, Boris Hanin, and Joel Hestness. Don’t be lazy: Completep enables compute-efficient deep transformers. arXiv preprint arXiv:2505.01618, 2025. URL https://doi.org/10.48550/arXiv.2505.01618.

---

> ### Author Response · Authors · 2025-11-26
> **Performance of Shampoo**
>
> $\text{Ans}3.$
>
> We thank the reviewer for bringing this point up.
> We would like to point out that besides the hyperparameters inherent to the optimization routine (LR, weight decay, betas, epsilon, etc), Shampoo has important parameters related to the preconditioners it uses. Those parameters were not tuned here and we are using default values which might explain the discrepancy. Furthermore, because of the linear algebra routines it uses, Shampoo benefits from the use of double precision arithmetic to be accurate while we are using half precision. See [1,2] for more details. Our goal here is to show stable transfer of the optimization routine hyperparameters.
>
> Additionally, the reviewer pointed out that the smallest width (128) has a lower loss than the higher widths for Shampoo. This is a very keen observation and it occurs because the maximal update guarantees hold asymptotically, i.e., as the model width goes to infinity. This implies that if the model width is not "large enough" then the expected limiting behavior will not be observed for shallow models. Interestingly, a similar behavior is observed for ADOPT and LAMB on NanoGPT, where the optimal learning rates for the model width of 128 is different than the optimal learning rates for larger widths. Under $\mu$P, the value of optimal learning rate gradually stabilizes, but only after the model width if larger than some threshold. We have discussed this behavior in more details in Appendix B.1. In the discussion we also emphasize that preliminary works have suggested to scale the $\mu$P parameterization by the width of a "base model", on which the hyperparameters will be tuned before transferring to the larger, target model. Rescaling $\mu$P using such as "base-model-width" might overcome the empirical discrepancies, but as we point out in our discussion, the "base-model-width" acts as an additional hyperparameter to $\mu$P. To avoid tuning this hyperparameter, we have fixed its value to $1$ in our experiments.
>
> REFERENCES:
>
> [1] Hao-Jun Michael Shi, Tsung-Hsien Lee, Shintaro Iwasaki, Jose Gallego-Posada, Zhijing Li, Kaushik Rangadurai, Dheevatsa Mudigere, and Michael Rabbat. A distributed data-parallel pytorch implementation of the distributed shampoo optimizer for training neural networks at-scale, 2023. URL https://arxiv.org/abs/2309.06497.
>
> [2] Nikhil Vyas, Depen Morwani, Rosie Zhao, Mujin Kwun, Itai Shapira, David Brandfonbrener, Lucas Janson, and Sham Kakade. Soap: Improving and stabilizing shampoo using adam. arXiv preprint arXiv:2409.11321, 2024.

---

### Official Review · Reviewer_6App · 2025-10-31

**Soundness:** 2
**Presentation:** 2
**Contribution:** 1
**Rating:** 4
**Confidence:** 3

**Summary:**

The paper recasts μP through a spectral-norm lens: enforce layerwise spectral scaling on weights and updates so that a single training step yields width-invariant functional change; then derive optimizer-specific LR rules. Using this recipe, the authors extend μP-style scaling from SGD/Adam to AdamW, ADOPT, LAMB, Sophia, and Shampoo, and demonstrate zero-shot LR transfer across width on NanoGPT and small-budget LLaMA-2; depth results are preliminary.

**Strengths:**

1. A clear, interpretable derivation that produces width-invariant HPs (esp. LR) for multiple optimizers, broadening μP’s practical coverage with lighter machinery than Tensor Programs.

2. Closed-form LR scalings plus explicit forward scaling and (O(1)) treatment for LN/bias—usable as a “cookbook.”

3. Multiple widths trained per model family: NanoGPT (128→2048) and LLaMA-2–style (256→2048 ≈154M→1.38B params on WikiText-103). LR–vs–loss sweeps show the same LR tuned on the smallest width remains optimal at larger widths.

4. Coordinate checks are stable; LR transfer across depth looks promising for AdamW/Sophia, highlighting where more theory is needed.

5. Coordinate-check plots + LR tables make the recipe easy to audit and adopt.

**Weaknesses:**

1.The analysis repackages μP using the published spectral condition and **retains μP’s assumptions**. Under the same assumptions, Tensor Programs could in principle obtain the same optimizer scalings. A genuine advance would relax assumptions or prove depth-scaling for the added optimizers.

2.Core derivations (Result 4.1) use a **linear MLP, batch-1** (rank-1 gradients where spectral≈Frobenius). Transformer attention is not newly analyzed—assumptions are imported and only **validated empirically**.

3.Width LR sweeps + coordinate checks substantially repeat the original μP methodology; runs are **small/medium-scale** (short LLaMA-2 training on WikiText-103), illustrating trends rather than establishing robustness at modern pretraining scales.

4.Clear oscillations/instabilities for ADOPT/LAMB/Shampoo with depth are reported but not diagnosed (e.g., trust-ratio dynamics, Shampoo preconditioner spectra, finite-width violations).

5.Non-monotone behaviors (LAMB/Shampoo) across width/depth lack error-budget analyses (e.g., per-layer update spectral norms vs target).

6.Since TP-V already covered Adam, scaling tables, and coordinate checks, the paper should more explicitly delineate **what is new** (optimizer catalog via spectral derivation) vs **what is inherited**.

**Questions:**

Which μP/spectral assumptions are actually essential? Can you prove width or depth transfer for any added optimizer under weaker/realistic settings (batches >1, attention+LN in the loop)?

Can you provide a formal spectral argument for self-attention (Q/K/V/O) under your update-norm rules (e.g., block-matrix bounds), rather than relying solely on experiments?

Please instrument per-layer **update spectral norms** and report **LAMB trust ratios / Shampoo spectra** across depth to explain the oscillations.

Add probes isolating finite-width corrections, batch-size effects, and residual-path scaling interactions to map where spectral sufficiency is tight vs loose.

---

> ### Author Response · Authors · 2025-11-26
> **Clarifying assumptions and applicability of proposed framework**
>
> We thank the reviewer for their insightful feedback regarding our work. Incorporating their valuable suggestions has improved the readability and clarity of our manuscript. Please see below for detailed answers.
>
> $\text{Ans}1.$
>
> We realize that the original manuscript didn't clearly specify which assumptions are necessary to derive the different scaling laws and which assumptions are required to use the scalings in implementation. Therefore, we have modified the original document by moving the assumptions from the Appendix to Section 3.3 and added insights justifying why the assumptions should hold in practice. All our derivations hold exactly for a linear MLP trained with batch size 1. The assumptions in Section 3.3 are required to justify why the derived scalings in Table 2 can be directly implemented for more complex and practically used models, including LLMs. The assumptions in Section 3.3 are true in practice because they are made on the norms of the relevant quantities and don't restrict the nature of activation functions, vectors, updates etc. significantly. We also provide reference to prior work which empirically validates some of the assumptions on batch sizes made in Section 3.3. While the assumptions on the model in Section 3 seem restrictive, they allow our derivations to hold precisely and prevent significant approximation error in the theoretical analysis itself. In fact, the supporting results in Remark 1 and Remark 2 hold exactly because of the rank $1$ nature of the update matrix, and these results are essential for the $\mu$P derivation for Shampoo. By eliminating large approximation errors from the analytical derivations, the assumptions in Section 3.3 solely focus on addressing how the derived $\mu$P scalings can be applied directly to more complex models used in practice. This systematic delineation will also aid future studies to relax some of the assumptions made in this work.
>
> While we don't relax the assumptions from prior work, the proposed framework can collectively derive the $\mu$P scaling laws for the different types of layers (input, output, hidden) in the model. This is in contrast to, for example [1], where tensor program approach is used and layers need to be treated differently. Additionally, under our systematic and tractable framework, we are able to derive the $\mu$P scalings for several different optimizers. In contrast, there are rarely any existing works which have successfully extended $\mu$P to other optimizers, primarily because of the intractable tensor program calculations. A notable exception is the work by [2] where the authors derived $\mu$P for Shampoo. However, in comparison to [2], we arrive at the same $\mu$P scalings for Shampoo by using the more tractable spectral conditions, thereby significantly reducing the burden of tedious calculations under the tensor programs approach as well as providing an easy way to extend $\mu$P to other second-order methods. The novel scalings for different optimziers are highlighted in Table 2, where the scalings in black are derived from the proposed spectral norm framework, and the scalings in red are derived in the original paper [1]. As you can see, we not only recover the scalings from the original work for AdamW, but also extend $\mu$P for ADOPT, Sophia, LAMB, Shampoo and Muon.
>
> REFERENCES:
>
> [1] Greg Yang, Edward Hu, Igor Babuschkin, Szymon Sidor, Xiaodong Liu, David Farhi, Nick Ryder, Jakub Pachocki, Weizhu Chen, and Jianfeng Gao. Tuning large neural networks via zero-shot hyperparameter transfer. Advances in Neural Information Processing Systems, 34:17084–17097, 2021. URL https://doi.org/10.48550/arXiv.2203.03466.
>
> [2] Satoki Ishikawa and Ryo Karakida. On the parameterization of second-order optimization effective towards the infinite width. In The Twelfth International Conference on Learning Representations.

---

> ### Author Response · Authors · 2025-11-26
> **Handling attention blocks**
>
> $\text{Ans}2.$
>
> We thank the reviewer for asking for further clarification on how the proposed framework can be applied to the attention blocks. To determine the scaling for the attention blocks, we use the spectral norm condition on the weight matrix, $W_l$, from (C.1). Let us define the attention block as the product of two matrices, i.e., $\text{Attention} = A * V$ where matrix $A\in \mathbb{R}^{d_{model} \times d_{model}}$ is the output of the softmax operation and matrix $V \in \mathbb{R}^{d_{model} \times {d_v n_{head}}}$. Since the output of the softmax operation is bounded by $1$, we can get a tight upper bound for the spectral norm as  $|| \text{Attention} || <= || V ||$. The spectral norm of $V$ will depend on its initialization. For now, if we assume that the elements of $V$ are initialized by standard normal distribution, then the spectral norm $|| V || \approx \Theta( \sqrt{d_{model}} + \sqrt{d_v n_{head}} )$ from results in random matrix theory [1,2]. Finally, to satisfy (C.1), the attention logits should be scaled by a factor $\Theta( 1 / \sqrt{d_{model}} \min [ 1, \sqrt{d_{model}} / \sqrt{d_v n_{head}} ] )$. Typically, $d_v n_{head} = d_{model}$, which implies that the attention logits should be scaled by an additional factor of $1/\sqrt{d_{model}}$. This scaling matches the results from literature [3]. We can also substitute $n_l = n_{l-1} = d_{model}$ in Table 2 to get the appropriate scalings for different optimizers for the attention block.
>
> REFERENCES:
>
> [1] Mark Rudelson and Roman Vershynin. Non-asymptotic theory of random matrices: extreme singular values. In Proceedings of the International Congress of Mathematicians 2010 (ICM 2010) (In 4 Volumes) Vol. I: Plenary Lectures and Ceremonies Vols. II–IV: Invited Lectures, pp. 1576–1602. World Scientific, 2010.
>
> [2] Roman Vershynin. High-dimensional probability: An introduction with applications in data science, volume 47. Cambridge university press, 2018.
>
> [3] Greg Yang, Edward Hu, Igor Babuschkin, Szymon Sidor, Xiaodong Liu, David Farhi, Nick Ryder, Jakub Pachocki, Weizhu Chen, and Jianfeng Gao. Tuning large neural networks via zero-shot hyperparameter transfer. Advances in Neural Information Processing Systems, 34:17084–17097, 2021. URL https://doi.org/10.48550/arXiv.2203.03466.

---

> ### Author Response · Authors · 2025-11-26
> **Clarifications on depth-scaling**
>
> $\text{Ans}3.$
>
> We thank the reviewer for their insightful suggestion regarding the oscillation analysis. In fact, the bottom row of coordinate check plots in Appendix B.2 plot the layerwise, relative mean values of the feature vectors under depth-scaling. The plots demonstrate stable behavior across increasing depth, implying that although maximizing feature learning is sufficient for hyperparameter transfer across model width, it is not sufficient to guarantee hyperparameter transfer across model depth. We further refer the reviewer to [1] where the authors suggest that maximizing feature diversity is the dominant factor in depth-scaling problems. Therefore, the conditions in (C.1) do not guarantee depth-scaling and so far, we are not aware of any existing work that encapsulates feature diversity in terms of spectral norms. Given that we are not aware of the limiting spectral norm behavior of the weight matrices under depth-scaling, perhaps the first direction for future work is to find the equivalent spectral conditions for depth-scaling and then use a generic framework as proposed in this work to derive the scalings for a general class of optimizers.
>
> $\text{Ans}4.$
>
> We thank the reviewer for their feedback. We appreciate the care taken while reading our work and providing us with thoughtful feedback. We would appreciate further clarification to address this question accurately. Is the reviewer asking for empirical demonstration of zero-shot transfer of batch-size?  Additionally, can the reviewer please elaborate on which values to report for "residual-path scaling interactions"?
>
> Regarding the query on finite-width corrections, we believe the reviewer is referring to scaling $\mu$P results with a "base-model-width". In Appendix B.1, we discuss our observation that adding the "base-model-width" factor acts as an additional hyperparameter to $\mu$P and requires tuning. To avoid this, we have fixed the "base-model-width" to $1$ in our simulations.
>
>
> REFERENCE:
>
> [1] Greg Yang, Dingli Yu, Chen Zhu, and Soufiane Hayou. Tensor programs vi: Feature learning in infinite-depth neural networks. arXiv preprint arXiv:2310.02244, 2023. URL https://doi.org/10.48550/arXiv.2310.02244.

---

### Meta-Review · Area_Chair_Ncp8 · 2025-12-30

**Summary:**

This paper proposes a framework to derive \mu P for a broader class of optimizers, including AdamW, ADOPT, LAMB, Sophia, Shampoo and Muon, building on recent work that introduced spectral conditions as an alternative to tensor programs. Numerical experiments on NanoGPT and Llama-2 are provided to validate the derived parameterization.

Reviewers have several major concerns, e.g., 1) no derivation for Muon optimizer; 2) Core derivations are not new; 3) several (simplifying) assumptions required for derivations; 4) the scalability of proposed method.

The authors gave detailed responses. They made clarifications on the issues of assumptions, and added derivations for Muon optimizer. Some reviewers may change their score slightly. But overall, the paper requires a major revision and reevaluation (e.g., the correctness of newly added derivations) for improvement.

**Reviewer Concerns:**

Reviewer 6App has major concerns: 1) The analysis repackages μP using the published spectral condition and retains μP’s assumptions; 2) Core derivations are not new; 3) Width LR sweeps + coordinate checks substantially repeat the original μP methodology; 4) The lack of error-budget analyses for non-monotone behaviors (LAMB/Shampoo) across width/depth; 5) the clear comparison with TP-V.

The authors gave detailed responses. They clarified some issues of assumptions and showed the applicability of proposed framework, but they also acknowledged that they don't relax the assumptions from prior work. They clarified on how the proposed framework can be applied to the attention blocks, and made explanations on depth-scaling.

Reviewer aLhn has major concerns: 1) no derivation for Muon optimizer; 2) the derivations rely on strong simplifications and the dropping of weight decay; 3) No clear zero-shot LR transfer for the results of Shampoo; 4) preliminary depth scaling.

The authors gave detailed responses. Particularly, they provided new derivations for Muon optimizer and discussed the derivations under other settings of the hyperparameters. If the new derivations are correct, Reviewer aLhn may increase the score. But clearly, the paper needs a major revision and reevaluation.

Reviewer gp6p has major concerns: 1) limited intellectual novelty of the methodology; 2) limited theoretical scope.

The authors made clarifications on contributions and assumptions required to move from theory to practice, added P derivation for Muon, and restructured Section 4. The authors’ responses may address some concerns, but I am afraid that the reviewer will not change the concern on the novelty.

Reviewer qLDZ has major concerns: 1) the scalability of proposed method; 2) simplifying assumptions; 3) requiring many assumptions for derivations; 4) no derivation for Muon.

The authors clarified the issues on the scalability of proposed method, and thought their experimental setting is reasonable. They also made clarifications on required assumptions and added the derivation for Muon.

**Reviewer Scores:**

Though the authors answered the questions raised by Reviewer QqGB, most weaknesses remain. Thus, I think Reviewer 6App may not change her/his score.

Reviewer aLhn may increase the score if the new derivations in the authors’ response are correct. But clearly, this also requires a major revision of the paper.

The authors’ responses may address some concerns of Reviewer gp6p, but I am afraid that the reviewer will not change the concern on the novelty, and thus keep the score unchanged.

I think Reviewer qLDZ may not change her/his score, as some weaknesses (e.g., the scalability of proposed method) have not been addressed directly.

---

### Decision · Program_Chairs · 2026-01-26

Reject